# The devil's in the disequilibrium: multi-component analysis of dissolved carbon and oxygen changes under a broad range of forcings in a general circulation model

Sarah Eggleston[1,*] and Eric D. Galbraith[1,2,3]

[1]Institut de Ciència i Tecnologia Ambientals (ICTA), Universitat Autònoma de Barcelona, 08193 Barcelona, Spain
[2]Institució Catalana de Recerca i Estudis Avançats (ICREA), Pg. Lluís Companys 23, 08010 Barcelona, Spain
[3]Department of Earth and Planetary Science, McGill University, Montreal, Quebec H3A 2A7, Canada
[*]Now at: Laboratory for Air Pollution & Environmental Technology, Empa, Überlandstrasse 129, 8600 Dübendorf, Switzerland

*Correspondence to:* S. Eggleston (sarah.eggleston@gmail.com)

**Abstract.** The complexity of dissolved gas cycling in the ocean presents a challenge for mechanistic understanding and can hinder model intercomparison. One helpful approach is the conceptualization of dissolved gases as the sum of multiple, strictly-defined components. Here we decompose dissolved inorganic carbon (DIC) into four components: saturation ($DIC_{sat}$), disequilibrium ($DIC_{dis}$), carbonate ($DIC_{carb}$) and soft tissue ($DIC_{soft}$). The cycling of dissolved oxygen is simpler, but can still be aided by considering $O_2$, $O_{2_{sat}}$ and $O_{2_{dis}}$. We explore changes in these components within a large suite of simulations with a complex coupled climate-biogeochemical model, driven by changes in astronomical parameters, ice sheets and radiative forcing, in order to explore the potential importance of the different components to ocean carbon storage on long timescales. We find that both $DIC_{soft}$ and $DIC_{dis}$ vary over a range of 40 $\mu$mol/kg in response to the climate forcing, equivalent to changes in atmospheric $p$CO$_2$ on the order of 50 ppm for each. The most extreme values occur at the coldest and intermediate climate states. We also find significant changes in $O_2$ disequilibrium, with large increases under cold climate states. We find that, despite the broad range of climate states represented, changes in global $DIC_{soft}$ can be quantitatively approximated by the product of deep ocean ideal age and the global export production flux. In contrast, global $DIC_{dis}$ is dominantly controlled by the fraction of the ocean filled by Antarctic Bottom Water (AABW). Because the AABW fraction and ideal age are inversely correlated among the simulations, $DIC_{dis}$ and $DIC_{soft}$ are also inversely correlated, dampening the overall changes in DIC. This inverse correlation could be decoupled if changes in deep ocean mixing were to alter ideal age independently of AABW fraction, or if independent ecosystem changes were to alter export and remineralization, thereby modifying $DIC_{soft}$. As an example of the latter, we show that iron fertilization causes both $DIC_{soft}$ and $DIC_{dis}$ to increase and that the relationship between these two components depends on the climate state. We propose a simple framework to consider the global contribution of $DIC_{soft} + DIC_{dis}$ to ocean carbon storage as a function of the surface preformed nitrate and $DIC_{dis}$ of dense water formation regions, the global volume fractions ventilated by these regions, and the global nitrate inventory.

# 1 Introduction

The controls on ocean carbon storage are not yet fully understood. Although potentially very important, given the large inventory of dissolved inorganic carbon (DIC) the ocean contains (currently 38,000 Pg C vs. 700 Pg C in the pre-industrial atmosphere), the nuances of carbon chemistry, the dependence of air-sea exchange on wind stress and sea ice cover, the intricacies of ocean circulation, and the activity of the marine ecosystem all contribute to making it a very complex problem. The scale of the challenge is such that, despite decades of work, the scientific community has not yet been able to satisfactorily quantify the role of the ocean in the natural variations of $p\mathrm{CO_2}$ between 180 and 280 ppm that occurred over ice age cycles. This failure reflects persistent uncertainty that also impacts our ability to accurately forecast future ocean carbon uptake (Le Quéré et al., 2007; Friedlingstein et al., 2014).

In order to help with process understanding, Volk and Hoffert (1985) proposed conceptualizing ocean carbon storage as consisting of a baseline surface-ocean average, enhanced by two "pumps" that transfer carbon to depth: the solubility pump, produced by the vertical temperature gradient, and the soft-tissue pump, produced by the sinking and downward transport of organic matter. This conceptualization proved very useful, but it fails to deal explicitly with the role of spatially- and temporally-variable air-sea exchange, and cannot effectively address changes in ocean circulation. A number of other conceptual systems have been employed (e.g. Broecker et al., 1985; Gruber et al., 1996), both for considering natural changes in the carbon cycle of the past and the anthropogenic transient input of carbon into the ocean.

Here, we use the decomposition laid out by Williams and Follows (2011), with the small change that we consider only DIC, rather than total carbon. This theoretical framework defines four components that, together, add up to the total DIC: saturation ($\mathrm{DIC_{sat}}$), disequilibrium ($\mathrm{DIC_{dis}}$), carbonate ($\mathrm{DIC_{carb}}$) and soft tissue ($\mathrm{DIC_{soft}}$) (Ito and Follows, 2013; Bernardello et al., 2014; Ödalen et al., 2018). The first two components are "preformed" quantities ($\mathrm{DIC_{pre}} = \mathrm{DIC_{sat}} + \mathrm{DIC_{dis}}$), i.e. they are defined in the surface layer of the ocean and are carried passively by ocean circulation into the interior. In contrast, the latter two are equal to zero in the surface layer and accumulate in the interior due to biogeochemical activity (see fig. 1). Note that the four components are diagnostic quantities only, intended to aid in understanding mechanisms and clarifying hypotheses, and do not influence the behavior of the model (although they can be calculated more conveniently by including additional ocean model tracers, as described in Methods).

Saturation DIC is simply determined by the atmospheric $\mathrm{CO_2}$ concentration and its solubility in seawater, which is a function of ocean temperature, salinity, and alkalinity. For example, cooling the ocean will increase $\mathrm{CO_2}$ solubility, thereby leading to an increase in $\mathrm{DIC_{sat}}$. Given known changes in temperature, salinity, alkalinity, and atmospheric $p\mathrm{CO_2}$, the effective storage of $\mathrm{DIC_{sat}}$ can be calculated precisely.

At the ocean surface, primary producers take up DIC. The organic carbon that is formed then sinks or is subducted (as dissolved or suspended organic matter) and is transformed into remineralized DIC within the water column (a small fraction is buried at depth). Here we define $\mathrm{DIC_{soft}}$ as that which accumulates through the net respiration of organic matter below the top layer of the ocean (in our model, the uppermost 10 m). Thus, $\mathrm{DIC_{soft}}$ depends both on the export flux of organic matter, affected by surface ocean conditions including nutrient supply (Moore et al., 2013; Martin, 1990), and the ocean circulation as a whole,

including the surface-to-deep export and the flushing rate of the deep ocean, which clears out accumulated $DIC_{soft}$ (Toggweiler et al., 2003). The Southern Ocean (SO) is thought to be an important region for such changes on glacial/interglacial timescales, as the ecosystem there is currently iron-limited, and it also plays a major role in deep ocean ventilation (Martin, 1990; François et al., 1997; Toggweiler et al., 2003; Köhler and Fischer, 2006; Sigman et al., 2010; Watson et al., 2015; Jaccard et al., 2016).

Assuming a constant global oceanic phosphate inventory and constant C:P ratio, $DIC_{soft}$ would be stoichiometrically related to the preformed $PO_4^{3-}$ ($PO_{4_{pre}}^{3-}$) inventory of the ocean, where $PO_{4_{pre}}^{3-}$ is the concentration of $PO_4^{3-}$ in newly-subducted waters, and a passively-transported tracer in the interior. The potential to use $PO_{4_{pre}}^{3-}$ as a metric of $DIC_{soft}$ prompted very fruitful efforts to understand how it could change over time (Ito and Follows, 2005; Marinov et al., 2008a; Goodwin et al., 2008), though it has been pointed out that the large variation in C:P of organic matter could weaken the relationship between $DIC_{soft}$ and $PO_{4_{pre}}^{3-}$

(Galbraith and Martiny, 2015). Given that the variability of N:C is significantly smaller than P:C (Martiny et al., 2013), we use preformed nitrate in the discussion below.

Similar to $DIC_{soft}$, $DIC_{carb}$ is defined here as the DIC generated by the dissolution of calcium carbonate shells below the ocean surface layer. Note that this does not include the impact that shell production has at the surface; calcification causes alkalinity to decrease in the surface ocean, raising surface $pCO_2$ and shifting carbon to the atmosphere. Rather, within the framework

used here, this effect on alkalinity distribution falls under $DIC_{sat}$, since it alters the solubility of DIC. This highlights an important distinction between the four-component framework, which strictly defines subcomponents of DIC, and the "pump" frameworks, which provide looser descriptions of vertical fluxes of carbon and, in some cases, alkalinity. Changes in $DIC_{carb}$ on the timescales of interest are generally thought to be small compared to those of $DIC_{sat}$ and $DIC_{soft}$.

Typically, only these three components are considered as the conceptual drivers behind changes in the air-sea partitioning

of $pCO_2$ (e.g., IPCC, 2007; Kohfeld and Ridgwell, 2009; Marinov et al., 2008a; Goodwin et al., 2008). However, a fourth component, $DIC_{dis}$, is also potentially significant as discussed by Ito and Follows (2013). Defined as the difference between preformed DIC and $DIC_{sat}$, $DIC_{dis}$ can be relatively large (Takahashi et al., 2009) because of the slow timescale of atmosphere-surface ocean equilibrium of carbon compared to other gases, caused by the reaction of $CO_2$ with seawater (e.g., Zeebe and Wolf-Gladrow, 2001; Broecker and Peng, 1974; Galbraith and Martiny, 2015). In short, $DIC_{dis}$ is a function of all non-

equilibrium processes on DIC in the surface ocean, including biological uptake, ocean circulation, and air-sea fluxes of heat, freshwater and $CO_2$.

Like $DIC_{sat}$, $DIC_{dis}$ is a conservative tracer determined in the surface ocean, with no sources or sinks in the ocean interior. Since the majority of the ocean is filled by water originating from small regions of the Southern Ocean and the North Atlantic, the net whole-ocean disequilibrium carbon is approximately determined by the $DIC_{dis}$ in these areas weighted by the fraction of

the ocean volume filled from each of these sites. Unlike the other three components, $DIC_{dis}$ could contribute either additional oceanic carbon storage ($DIC_{dis} > 0$) or reduced oceanic carbon storage ($DIC_{dis} < 0$). Although this parameter is implicitly included in most models, studies using preformed nutrients as a metric for biological carbon storage have often ignored the potential importance of $DIC_{dis}$ by assuming fast air-sea gas exchange (e.g., Marinov et al., 2008a; Ito and Follows, 2005). In the pre-industrial ocean this is of little importance, given that global $DIC_{dis}$ is small because the opposing effects of North

Atlantic and Antarctic water masses largely cancel each other. However, Ito and Follows (2013) showed that $DIC_{dis}$ can have

a large impact by amplifying changes in $DIC_{soft}$ under constant pre-industrial ocean circulation, and Ödalen et al. (2018) have very recently shown that $DIC_{dis}$ can vary significantly in response to changes in ocean circulation states.

The cycling of dissolved oxygen is simpler than DIC. Because $O_2$ does not react with seawater or the dissolved constituents thereof, it has no dependence on alkalinity, and its equilibration with the atmosphere through air-sea exchange occurs approximately one order of magnitude faster than DIC (on the order of one month, rather than one year). Nonetheless, dissolved oxygen can be conceptualized as including a preformed component, which is the sum of saturation and disequilibrium, and an oxygen utilization component, which is given by the difference between the in situ and preformed $O_2$ in the ocean interior. Apparent Oxygen Utilization (AOU), typically taken as a measure of accumulated respiration, can be misleading if the preformed $O_2$ concentration differed significantly from saturation, i.e. if $O_{2_{dis}}$ is significant, as it appears to be in high latitude regions of dense water formation (Ito et al., 2004; Duteil et al., 2013; Russell and Dickson, 2003). If $O_{2_{dis}}$ varies with climate state, it might contribute significantly to past or future oxygen concentrations.

Here, we use a complex Earth system model to investigate the potential changes in the constituents of DIC and $O_2$ on long timescales, relevant for past climate states as well as the future. We make use of a large number of equilibrium simulations, conducted over a wide range of radiative, orbital and ice sheet boundary conditions, as a "library" of contrasting ocean circulations in order to test the response of disequilibrium carbon storage to physically plausible changes in ocean circulation. The basic physical aspects of these simulations were described by Galbraith and de Lavergne (2018). We supplement these with a smaller number of iron fertilization experiments to examine the additional impact of circulation-independent ecosystem changes. In order to simplify the interpretation, we chose to prescribe a constant $pCO_2$ of 270 ppm for the air-sea exchange in all simulations, as also done by Ödalen et al. (2018). Thus, the changes in $DIC_{sat}$ reflect only changes in temperature, salinity, alkalinity and ocean circulation arising from the climate response, and not changes in $pCO_2$. Nor do they explicitly consider changes in the total carbon or alkalinity inventories driven by changes in outgassing and/or burial (Roth et al., 2014; Tschumi et al., 2011); rather, the alkalinity inventory is fixed, and the carbon inventory varies due to changes in total ocean DIC (since the atmosphere is fixed). As such, the experiments here should be seen as idealized climate-driven changes, and should be further tested with more comprehensive models including interactive $CO_2$.

## 2  Methods

### 2.1  Model description

The GCM used in this study is CM2Mc, the Geophysical Fluid Dynamics Laboratory's Climate Model version 2 but at lower resolution (3º), described in more detail by Galbraith et al. (2011) and modified as described by Galbraith and de Lavergne (2018). This includes the Modular Ocean Model version 5, a sea ice module, static land, and static ice sheets, and a module of Biogeochemistry with Light, Iron, Nutrients and Gases (BLINGv1.5) (Galbraith et al., 2010). Unlike BLINGv0, BLINGv1.5 allows for variable P:C stoichiometry using the "Line of Frugality" (Galbraith et al., 2015) and calculates the mass balance of phytoplankton in order to prevent unrealistic bloom magnitudes at high latitudes, reducing the magnitude of disequilibrium $O_2$, which was very high in BLINGv0 (Duteil et al., 2013; Tagliabue et al., 2016). In addition, three water mass tracer tags are

defined: a Southern tracer south of 30ºS, a North Atlantic tracer north of 30ºN in the Atlantic, and a North Pacific tracer north of 30ºN in the Pacific. An "ideal age" tracer is also defined as zero in the global surface layer, and increasing in all other layers at a rate of 1 year per year.

## 2.2 Experimental design

The model runs analyzed here are part of the same suite of simulations discussed by Galbraith and de Lavergne (2018). A control run was conducted with a radiative forcing equivalent to 270 ppm atmospheric $pCO_2$ and the Earth's obliquity and precession set to modern values (23.4º and 102.9º, respectively). Experimental simulations were run at values of obliquity (22º, 24.5º) and precession (90º, 270º) representing the astronomical extremes encountered over the last 5 My (Laskar et al., 2004), while eccentricity was held constant at 0.03. A range of greenhouse radiative forcings was imposed equivalent to $pCO_2$

levels of 180, 220, 270, 405, 607, or 911 ppm; with reference to a pre-industrial radiative forcing, the radiative forcings are roughly equal to $-2.2$, $-1.1$, $0$, $+2.2$, $+4.3$, and $+6.5$ W/m$^2$, respectively (Myhre et al., 1998).

The biogeochemical component of the model calculates air-sea carbon fluxes using a fixed atmospheric $pCO_2$ of 270 ppm throughout all model runs. Note that 270 ppm was chosen to reflect an average interglacial level, rather than specifically focusing on the pre-industrial climate state. This use of constant $pCO_2$ for the carbon cycle means that the DIC$_{sat}$ is not

consistent with the $pCO_2$ used for the radiative forcing, so that changes in DIC$_{sat}$ caused by a given $pCO_2$ change would tend to be larger than would be expected in reality. This has a negligible effect on the other carbon components, given that they do not depend directly on $pCO_2$; this has been confirmed by model runs with the University of Victoria Earth System Model using a similar decomposition strategy (Khatiwala et al., *in prep.*).

Eight additional runs were conducted using Last Glacial Maximum (LGM) ice sheets with the lowest two radiative forcings

and the same orbital parameters. Iron fertilization simulations calculate the input flux of dissolved iron to the ocean surface assuming a constant solubility and using the glacial atmospheric dust field of Mahowald et al. (2006) as modified by Nickelsen and Oschlies (2015) instead of the standard pre-industrial dust field; note that this is not entirely in agreement with more modern reconstructions, which could potentially have an influence on the induced biological blooms, both in magnitude and geographically (e.g. Albani et al., 2012, 2016). Four iron fertilization experiments were run with the lowest radiative forcing

with LGM ice sheets, as well as one model run similar to the control run. Finally, two simulations were run that were identical to the pre-industrial setup, but the rate of remineralization of sinking organic matter is set to 75% of the default rate, approximately equivalent to the expected change due to a 5ºC ocean cooling (Matsumoto et al., 2007); one of these runs also includes iron fertilization. All simulations are summarized in Table 1.

In the following, three particular runs are highlighted for comparison to illustrate cold (CW), moderate (MW), and hot (HW)

worlds. These include radiative forcings of $-2.2$, $0$, and $+4.3$ W/m$^2$, respectively; the former includes LGM ice sheets; and the obliquity and precession is 22º and 90º for GL and 22º and 270º for PI and WP. These specific runs are distinguished from glacial-like (GL) and interglacial-like (IG) scenarios, which refer to averages of four runs each, each with a radiative forcing of $-2.2$ and $0$ W/m$^2$, respectively; the GL runs also have LGM ice sheets.

All simulations were run for 2100 – 6000 model years beginning with a pre-industrial spinup. While the model years presented here largely reflect runs after having reached steady state, it is important to note that the pre-industrial run (41 in Table 1) still has a drift of 1 $\mu$mol/kg over the 100 y shown here and thus may not yet be at steady state.

## 2.3 Decompositions

The four-component DIC scheme and three-component $O_2$ scheme described in the introduction can be exactly calculated for any point in an ocean model using five easily-implemented ocean model prognostic tracers: DIC, $DIC_{pre}$, $DIC_{sat}$, $O_2$ and $O_{2pre}$, since: $DIC_{dis} = DIC_{pre} - DIC_{sat}$, $DIC_{soft} = (O_{2pre} - O_2) \cdot r_{C:O_2}$, $DIC_{dis} = DIC - DIC_{pre} - DIC_{soft}$, and $O_{2dis} = O_{2pre} - O_{2sat}$ ($O_{2sat}$ can be accurately calculated from the conservative tracers temperature and salinity). In the absence of a $DIC_{sat}$ tracer, it can be estimated from $alk_{pre}$, temperature and salinity. For this large suite of simulations, we only had DIC, alk, $O_2$ and $O_{2pre}$ available.

Thus, although we could calculate $O_{2dis}$ and $DIC_{soft}$ directly, it was necessary to use an indirect method to calculate the other three carbon tracers. Following Bernardello et al. (2014), we first estimate $alk_{pre}$ using a regression, then calculate $DIC_{sat}$ using $alk_{pre}$, temperature, salinity, and the known atmospheric $p$CO$_2$ of 270 ppm. $DIC_{carb}$ is calculated as $0.5 \cdot (alk - alk_{pre})$, and $DIC_{dis}$ is then calculated as a residual. For more details and an estimate of the error in the method, see the appendix.

## 3   Results and Discussion

### 3.1   General climate response to forcings

Differences in the general ocean state among the model simulations are described in detail by Galbraith and de Lavergne (2018). We provide here a few key points, important for interpreting the dissolved gas simulations. First, the AABW fraction of the global ocean varies over a wide range among the simulations, with abundant AABW under low and high global average Surface Air Temperature (SAT), and a minimum at intermediate SAT (fig. 2(b)). The NADW fraction is approximately the

inverse of this. The ventilation rate of the global ocean is roughly correlated with the AABW fraction, with rapid ventilation (small average ideal age) when the AABW fraction is high (fig. 2). The density difference between AABW and NADW is the overall driver of the AABW fraction and ventilation rate in the model simulations. The density difference can be explained, in all simulations colder than present-day, by the effect of sea ice cycling in the Southern Ocean on the global distribution of salt: when the Southern Ocean is cold and sea ice formation abundant, there is a large net transport of freshwater out of the

circum-Antarctic, causing AABW to become more dense. NADW becomes consequently fresher, because there is less salt left to contribute to NADW. As noted by Galbraith and de Lavergne (2018), the simulated ventilation rates should be viewed with caution, given the poor mechanistic representation of diapycnal mixing in the deep ocean, and the AABW fraction may not be correlated with ventilation rate in the real ocean.

## 3.2 General changes in DIC

Total DIC generally decreases from cold to warm simulations, under the constant $pCO_2$ of 270 ppm used for air-sea exchange. Changes in $DIC_{sat}$ drive the largest portion of this trend, decreasing approximately linearly with surface air temperature due to the temperature-dependence of $CO_2$ solubility, resulting in a difference of 50 $\mu$mol/kg over this range (see fig. 3). Because of the constant $pCO_2$ of 270 ppmv in the biogeochemical module, the simulated range of $DIC_{sat}$ is larger than would occur if the prescribed biogeochemical $pCO_2$ were the same as that used to produce the low SAT (Ödalen et al., 2018). $DIC_{carb}$ is relatively small in magnitude and generally increases with SAT, but has a standard deviation of only 4 $\mu$mol/kg over all simulations, so we do not discuss it further.

In contrast to $DIC_{sat}$ and $DIC_{carb}$, $DIC_{dis}$ and $DIC_{soft}$ vary nonlinearly with global temperatures, with a clear and shared turning point near the middle of the temperature range. Thus, the extreme values of $DIC_{soft}$ and $DIC_{dis}$ occur under the coldest state and at an intermediate state close to the pre-industrial. Both $DIC_{dis}$ and $DIC_{soft}$ are strongly correlated with ocean ventilation, quantified here by the global average of the ideal age tracer ($r^2$ = 0.69 and 0.89, respectively), and thus with each other ($r^2$ = 0.74). However, whereas Ito and Follows (2013) found a positive correlation of $DIC_{soft}$ and $DIC_{dis}$ under nutrient depletion experiments with constant climate, these experiments indicate that, when driven by the wide range of physical changes explored here, $DIC_{soft}$ and $DIC_{dis}$ are negatively correlated.

Simulated changes in $DIC_{dis}$ are of the same magnitude as the $DIC_{soft}$ changes, to which much greater attention has been paid. For a global average buffer factor between 8 and 14 (Zeebe and Wolf-Gladrow, 2001), a rough, back-of-the-envelope calculation shows that a 1 $\mu$mol/kg change in DIC corresponds to a 0.9 – 1.6 ppm change in atmospheric $pCO_2$ based on a DIC concentration of 2300 $\mu$mol/kg and $pCO_2$ of 270 ppm. Thus, the increase in the global average $DIC_{dis}$ in these simulations could have contributed more than a 40 ppm change in the atmospheric $pCO_2$ stored in the ocean during the glacial compared to today. It is important to recognize that the drawdown of $CO_2$ by disequilibrium storage would have resulted in a decrease of $DIC_{sat}$, given the dependence of the saturation concentration on $pCO_2$, so this estimate should not be interpreted as a straightforward atmospheric $pCO_2$ change. Nonetheless, while this is only a first-order approximation and the model biases are potentially large, it seems very likely that the disequilibrium carbon storage was a significant portion of the net 90 ppm difference.

## 3.3 Climate-driven changes in $DIC_{soft}$

The biogeochemical model used here is relatively complex, with limitation by three nutrients (N, P and Fe), denitrification and $N_2$ fixation, in addition to the temperature- and light-dependence typical of biogeochemical models. The climate model is also complex, including a full atmospheric model, a highly-resolved dynamic ocean mixed layer, and many nonlinear subgridscale parameterizations, and uses short (< 3 h) timesteps. The simulations we show span a wide range of behaviours, including major changes in ocean ventilation pathways and patterns of organic matter export.

Thus, it is perhaps surprising that the net global result of the biological pump, as quantified by $DIC_{soft}$, has highly predictable behavior. As shown in fig. 4, the global $DIC_{soft}$ varies closely with the product of the global average sinking flux of organic matter at 100 m and the average ideal age of the global ocean. Qualitatively this is not a surprise, given that greater export

pumps more organic matter to depth, and a large age provides more time for respired carbon to accumulate within the ocean. But the quantitative strength of the relationship is striking. As demonstrated in fig. 4, global $DIC_{soft}$ is not as well correlated with either of these parameters separately as it is with their produce, "age $\times$ export."

It is difficult to assess the likelihood that the real ocean follows this relationship to a similar degree. One reason it might differ is if remineralization rates vary spatially, or with climate state. In the model here, as in most biogeochemical models, organic matter is respired according to a globally-uniform power law relationship vs. depth (Martin et al., 1987). Kwon et al. (2009) showed that ocean carbon storage is sensitive to changes in these remineralization rates, and this would provide an additional degree of freedom. It is not currently known how much remineralization rates can vary naturally; they may vary as a function of temperature (Matsumoto et al., 2007) or ecosystem structure. As a result, the relationship between $DIC_{soft}$ and age $\times$ export may be stronger in the model than in the real ocean.

Nonetheless, the results suggest that, as a useful first-order approximation, the global change in $DIC_{soft}$ between two states can be given by a simple linear regression:

$$\Delta DIC_{soft}[\mu mol\ kg^{-1}] = m_1 \cdot \Delta(age[y] \times export[Pg\ C\ y^{-1}]) \tag{1}$$

or in terms of $pCO_2$:

$$\Delta pCO_{2,soft}[ppm] \approx m_2 \cdot \Delta(age[y] \times export[Pg\ C\ y^{-1}]) \tag{2}$$

Note that $m_2$ is a function of the buffer factor and the climate state (atmospheric $pCO_2$ and DIC). Based on the results here, $m_1 = 0.036$ and $m_2 = 0.065, 0.042, 0.029$ for modern (405 ppm $pCO_2$), pre-industrial (270 ppm) and glacial (180 ppm) conditions, respectively. Note that we have not varied $pCO_2$ in these simulations, so these equations are only meant to illustrate the mathematical relationship observed in fig. 4. This simple meta-model may provide a useful substitute for full ocean-ecosystem calculations, and should be further tested against other ocean-ecosystem coupled models with interactive $CO_2$. Note that, as for the disequilibrium estimate above, the soft tissue pump $CO_2$ drawdown would be partially compensated by a decrease in saturation carbon storage, so this will be larger than the net atmospheric effect. In addition, we have not accounted for consequent changes in the surface ocean carbonate chemistry (including changes in the buffer factor).

It is important to point out that the simulated change in $DIC_{soft}$ between interglacial and glacial states appears to be in conflict with reconstructions of the LGM. Proxy records appear to show that LGM dissolved oxygen concentrations were lower throughout the global ocean, with the exception of the North Pacific, implying greater $DIC_{soft}$ concentrations during the glacial then during the Holocene (Galbraith and Jaccard, 2015). In contrast, the model suggests that greater ocean ventilation rates in the glacial state (fig. 2(a)) would have led to reduced global $DIC_{soft}$. As discussed by Galbraith and de Lavergne (2018), radiocarbon observations imply that the model ideal age is approximately 200 y too young under glacial conditions compared to the LGM, suggesting a circulation bias that may reflect incorrect diapycnal mixing or non-steady-state conditions. Whatever the cause, if we take this 200 y bias into account, the regression implies an additional 33 $\mu mol\ kg^{-1}$ $DIC_{soft}$ were stored in the glacial ocean. This would bring the simulated glacial $DIC_{soft}$ close to, but still less than, the simulated pre-industrial value.

We propose that the apparent remaining shortfall in simulated glacial $DIC_{soft}$ could reflect one or more of the following non-exclusive possibilities: 1. the model does not capture changes in remineralization rates caused by ecosystem changes; 2.

the model underestimates the glacial increase in the nitrate inventory and/or growth rates, perhaps due to changes in the iron cycle; 3. the ocean was not in steady state during the LGM, and therefore not directly comparable to the GL simulation; 4. the inference of $DIC_{soft}$ from proxy oxygen records is incorrect due to significant changes in preformed oxygen disequilibrium (see below). If either of the first two possibilities is important, it would imply an inaccuracy in the meta-model derived here.

## 3.4 Climate-driven changes in $DIC_{dis}$

The ocean basins below 1 km depth are largely filled by surface waters subducted to depth in regions of deepwater formation (Gebbie and Huybers, 2011). In our simulations, water originating in the surface North Atlantic, termed NADW, and the Southern Ocean, termed AABW, make up 80-96% of this total deep ocean volume. Thus, to first order, the deep average $DIC_{dis}$ concentration can be approximated by a simple mass balance:

$$DIC_{dis_{deep}} \approx f_{AABW} \cdot DIC_{dis_{AABW}} + (1 - f_{AABW}) \cdot DIC_{dis_{NADW}} \tag{3}$$

Here, $f_{AABW}$ represents the fraction of deepwater originating in the SO, and $DIC_{dis_{AABW}}$, $DIC_{dis_{NADW}}$ represent the $DIC_{dis}$ concentrations at the sites of deepwater formation (see fig. 5). North Atlantic deep waters form with negative $DIC_{dis}$, reflecting surface undersaturation, while the Southern Ocean is supersaturated ($DIC_{dis} > 0$). These opposing tendencies between NADW and AABW cause a partial cancellation of $DIC_{dis}$ when globally averaged, which makes the disequilibrium component small in the modern ocean. Theoretically, the simulated $DIC_{dis}$ could change either due to changes in $f_{AABW}$ or the end-member compositions. Although the exact values of $DIC_{dis}$ in the two polar oceans vary among the simulations in response to climate (the reasons for which are discussed in more detail below), these changes are small relative to the consistent large contrast between $f_{AABW}$ and $f_{NADW}$, so that deep $DIC_{dis}$ is strongly controlled by the global balance of AABW vs. NADW in each simulation (see fig. 6). Global $DIC_{dis}$ becomes much larger when $f_{AABW}$ is larger, similar to the dynamic evoked by Skinner (2009). This is also illustrated by the depth transects of $DIC_{dis}$ in fig. 7.

We estimated the concentration of $DIC_{dis}$ in the regions of AABW and NADW formation, shown in fig. 5(b). The end members vary less significantly than $f_{AABW}$ over the range of simulations, in part due to competing effects of different processes. As discussed in section 3.1, simulations at both the low and high radiative forcing values used show increased AABW production, with a minimum at intermediate values (fig. 5). The fact that the highest $DIC_{dis_{AABW}}$ occurs at low SAT can be attributed to the rapid formation rate of AABW, while the intermediate-SAT minimum in AABW volume explains the minimum in global ocean $DIC_{dis}$ (fig. 3). We note that expanded terrestrial ice sheets shift the ratio of AABW to NADW to higher values, due to their impact on NADW temperature and downstream expansion of Southern Ocean sea ice (Galbraith and de Lavergne, 2018), further increasing $DIC_{dis}$ in glacial-like conditions. Sea ice in the Southern Ocean would be expected to exert a further control over $DIC_{dis}$, as this reduces air-sea gas exchange, thus allowing carbon to accumulate beneath the ice. However we did not perform experiments to isolate this effect.

## 3.5 Climate-driven changes in $O_{2_{dis}}$

Similarly to $DIC_{dis}$, $O_{2_{dis}}$ is defined as the departure from equilibrium of $O_2$ in the surface ocean with respect to the atmosphere and is advected into the ocean as a conservative tracer. Unlike C, $O_2$ does not react with seawater, and is present only as dissolved diatomic oxygen. Thus, $O_2$ has a much shorter time scale of exchange at the ocean-atmosphere interface, equilibrating one order of magnitude faster than $CO_2$. As a result, it is not sensitive to sea ice as long as there remains a fair degree of open water (Stephens and Keeling, 2000). But as the sea ice concentration approaches complete coverage, $O_2$ equilibration rapidly becomes quite sensitive to sea ice. If there is a significant undersaturation of $O_2$ in upwelling waters, the disequilibrium can become quite large (fig. 8).

In the model simulations, the $O_{2_{dis}}$ in the Southern Ocean becomes as large as $-100 \mu mol/kg$ in the coldest states (note that $DIC_{dis}$ and $O_{2_{dis}}$ are often anti-correlated). Because the disequilibrium depends on the $O_2$ depletion of waters upwelling at the Southern Ocean surface, this could potentially be even larger, if upwelling waters had lower $O_2$. We do not place a large degree of confidence in these values, given the likely sensitivity to poorly-resolved details of sea ice dynamics (e.g. ridging, leads) and dense water formation. Nonetheless, the potential for very large disequilibrium oxygen under cold states prompts the hypothesis that very extensive sea ice cover over most of the exposure pathway in the Southern Ocean might have made a significant contribution to the low $O_2$ concentrations reconstructed for the glacial (Jaccard et al., 2016; Lu et al., 2015).

## 3.6 Iron fertilization experiments

In addition to the changes driven by ice sheets, orbital and radiative forcing, we conducted iron fertilization experiments under glacial and pre-industrial-like conditions, including a simulation with reduced remineralization rates (fig. 9). As expected, both the global export and $DIC_{soft}$ increase when iron deposition is increased. However, the $DIC_{soft}$ increase is significantly lower in the well-ventilated GL simulations (2.9 $\mu mol/kg$) compared to the PI simulation (7.3 $\mu mol/kg$). This difference is qualitatively in accordance with the age $\times$ export relationship (fig. 4), though with a smaller increase of $DIC_{soft}$ than would be expected from the export increase, compared to the broad spectrum of climate-driven changes. This reduced sensitivity of $DIC_{soft}$ to the global export can be attributed to the fact that the iron-enhanced export occurs in the Southern Ocean, presumably because the remineralized carbon can be quickly returned to the surface by upwelling when ventilation is strong. Thus, the impact of iron fertilization on $DIC_{soft}$ is strongly dependent on Southern Ocean circulation.

The iron addition also causes an increase of $DIC_{dis}$ of approximately equal magnitude to $DIC_{soft}$ in the PI simulation and of relatively greater proportion in the GL simulations. Because the ocean in the GL simulations is strongly ventilated, the increase in export leads to an increase of $DIC_{dis}$, as remineralized $DIC_{soft}$ is returned to the Southern Ocean surface, where it has a relatively short residence time that limits outgassing to the atmosphere. With rapid Southern Ocean circulation, a good deal of the DIC sequestered by iron fertilization ends up in the form of $DIC_{dis}$, rather than $DIC_{soft}$ as might be assumed. Thus, just as the glacial state has a larger general proportion of $DIC_{dis}$ compared to $DIC_{soft}$, the iron addition under the glacial state produces a larger fraction of $DIC_{dis}$ relative to $DIC_{soft}$. The experiment in which the remineralization rate was reduced by 25% indicates that the effects of iron fertilization alone on both $DIC_{soft}$ and $DIC_{dis}$ are quite insensitive to the remineralization rate

(see fig. 9); for total DIC as well as for each component, the difference between the iron fertilization run and the corresponding control run in similar in panels (a) and (b). While we have not run a simulation under glacial-like conditions with a reduced remineralization rate, this suggests that the effects of iron fertilization and changes in the remineralization rate can be well-approximated as being linearly additive in this model.

The tendency to sequester carbon as $DIC_{dis}$ vs. $DIC_{soft}$, in response to iron addition, can be quantified by the global ratio $\Delta DIC_{dis}/\Delta DIC_{soft}$. Our experiments suggest that this ratio is 0.9 for the pre-industrial state and 3.3 for the glacial-like state. Because of the circulation dependence of this ratio, it is expected that there could be significant variation between models. It is worth noting that Parekh et al. (2006) found $\Delta DIC_{dis}/\Delta DIC_{soft}$ of 2 in response to iron fertilization, using a modern ocean circulation, as analyzed by Ito and Follows (2013). We also note that the quantitative values of $DIC_{soft}$ and $DIC_{dis}$ resulting from the altered iron flux should be taken with a grain of salt, given the very large uncertainty in iron cycling models (Tagliabue et al., 2016).

These results provide a note of caution for interpreting iron fertilization model experiments, which might be assumed to act primairly on the soft tissue carbon storage. At moderate ventilation rates (the pre-industrial control run), an increase in iron results in an increase both in $DIC_{soft}$, due to higher biological export, and $DIC_{dis}$, because of the upwelling of C-rich water resulting from higher remineralization, the effect discussed by Ito and Follows (2013). However, under a high ventilation state there is only a small increase in $DIC_{soft}$ in response to increased Fe in the surface ocean, as the remineralized carbon is quickly returned to the surface, thus producing a significant increase in $DIC_{dis}$ only. Thus, the carbon storage resulting from Southern Ocean iron addition could actually be dominated by $DIC_{dis}$ under some climate states, so that the overall impact may be significantly larger than would be predicted from $DIC_{soft}$ and/or $O_2$ utilization.

## 3.7 A unified framework for $DIC_{dis}$ and preformed nutrients

The concept of preformed nutrients allowed the production of a very useful body of work, striving for simple predictive principles. This work highlighted the importance of the nutrient concentrations in polar oceans where deep waters form (Sigman and Boyle, 2000; Ito and Follows, 2005; Marinov et al., 2008a) as well as changes in the ventilation fractions of AABW and NADW, given their very different preformed nutrient concentrations (Schmittner and Galbraith, 2008; Marinov et al., 2008b). Although the variability of P:C ratios implies significant uncertainty for the utility of $PO_{4_{pre}}^{3-}$ in the ocean, the relative constancy of N:C ratios suggests that $NO_{3_{pre}}^{-}$ is indeed linked to $DIC_{soft}$, inasmuch as the global N inventory is fixed (Galbraith and Martiny, 2015).

However, as shown by the analyses here, $DIC_{soft}$ – reflected by the preformed nutrients – is only half the story. Changes in $DIC_{dis}$ can be of equivalent magnitude, and can vary independently of $DIC_{soft}$ as a result of changes in ocean circulation and sea ice. Nonetheless, we find that the same conceptual approach developed for $DIC_{soft}$ can be used to predict $DIC_{dis}$ from the end member $DIC_{dis}$ and the global volume fractions. The preformed relationships and $DIC_{dis}$ can therefore be unified as follows (see fig. 10):

$$DIC_{soft} = NO_{3_{rem}}^{-} \cdot r_{C:N} \tag{4}$$

Remineralized nitrate can be expressed in terms of the global nitrate inventory and the accumulated nitrate loss due to pelagic and benthic denitrification:

$$\text{NO}_{3_{\text{rem}}}^- = \text{NO}_{3_{\text{global}}}^- - \text{NO}_{3_{\text{pre}}}^- + \text{NO}_{3_{\text{den}}}^- \tag{5}$$

$$\text{NO}_{3_{\text{pre}_{\text{upper}}}}^- \approx f_{\text{SO,upper}} \cdot \text{NO}_{3_{\text{pre}_{\text{SO,upper}}}}^- + f_{\text{NAtl,upper}} \cdot \text{NO}_{3_{\text{pre}_{\text{NAtl,upper}}}}^- + f_{\text{NPac,upper}} \cdot \text{NO}_{3_{\text{pre}_{\text{NPac,upper}}}}^- \tag{6}$$

$$\text{NO}_{3_{\text{pre}_{\text{deep}}}}^- \approx f_{\text{AABW}} \cdot \text{NO}_{3_{\text{pre}_{\text{AABW}}}}^- + f_{\text{NADW}} \cdot \text{NO}_{3_{\text{pre}_{\text{NADW}}}}^- \tag{7}$$

Because there is production of intermediate water but no deep convection in the North Pacific, we calculate this mass balance for the upper ocean (above 1 km) and deep ocean separately, dropping the Pacific Ocean term in eq. 7 for the deep ocean. For brevity, we continue with the derivation for the deep ocean only; the upper ocean follows analogously.

$$\text{DIC}_{\text{soft}_{\text{deep}}} \approx r_{\text{C:N}} \cdot [\text{NO}_{3_{\text{deep}}}^- + \text{NO}_{3_{\text{den,deep}}}^- - (f_{\text{AABW}} \cdot \text{NO}_{3_{\text{pre}_{\text{AABW}}}}^- + f_{\text{NADW}} \cdot \text{NO}_{3_{\text{pre}_{\text{NADW}}}}^-)] \tag{8}$$

Combining with equation 3,

$$\begin{aligned}
\text{DIC}_{\text{dis}_{\text{deep}}} + \text{DIC}_{\text{soft}_{\text{deep}}} \approx &\, r_{\text{C:N}} \cdot (\text{NO}_{3_{\text{deep}}}^- + \text{NO}_{3_{\text{den,deep}}}^-) + f_{\text{AABW}} \cdot (\text{DIC}_{\text{dis}_{\text{AABW}}} - r_{\text{C:N}} \cdot \text{NO}_{3_{\text{pre}_{\text{AABW}}}}^-) \\
&+ f_{\text{NADW}} \cdot (\text{DIC}_{\text{dis}_{\text{NADW}}} - r_{\text{C:N}} \cdot \text{NO}_{3_{\text{pre}_{\text{NADW}}}}^-)
\end{aligned} \tag{9}$$

Finally, the global average is computed by summing the volume-weighted values in the upper and deep ocean:

$$\text{DIC}_{\text{dis}_{\text{global}}} + \text{DIC}_{\text{soft}_{\text{global}}} \approx [V_{\text{upper}} \cdot (\text{DIC}_{\text{dis}_{\text{upper}}} + \text{DIC}_{\text{soft}_{\text{upper}}}) + V_{\text{deep}} \cdot (\text{DIC}_{\text{dis}_{\text{deep}}} + \text{DIC}_{\text{soft}_{\text{deep}}})]/V_{\text{total}} \tag{10}$$

Fully expanded, this yields:

$$\begin{aligned}
\text{DIC}_{\text{dis}_{\text{global}}} + \text{DIC}_{\text{soft}_{\text{global}}} \approx &\, \frac{V_{\text{upper}}}{V_{\text{total}}}[r_{\text{C:N}} \cdot (\text{NO}_{3_{\text{upper}}}^- + \text{NO}_{3_{\text{den,upper}}}^-) + f_{\text{SO,upper}} \cdot (\text{DIC}_{\text{dis}_{\text{SO,upper}}} - r_{\text{C:N}} \cdot \text{NO}_{3_{\text{pre}_{\text{SO,upper}}}}^-) \\
&+ f_{\text{NAtl,upper}} \cdot (\text{DIC}_{\text{dis}_{\text{NAtl,upper}}} - r_{\text{C:N}} \cdot \text{NO}_{3_{\text{pre}_{\text{NAtl,upper}}}}^-)] \\
&+ \frac{V_{\text{deep}}}{V_{\text{total}}}[r_{\text{C:N}} \cdot (\text{NO}_{3_{\text{deep}}}^- + \text{NO}_{3_{\text{den,deep}}}^-) + f_{\text{AABW}} \cdot (\text{DIC}_{\text{dis}_{\text{AABW}}} - r_{\text{C:N}} \cdot \text{NO}_{3_{\text{pre}_{\text{AABW}}}}^-) \\
&+ f_{\text{NADW}} \cdot (\text{DIC}_{\text{dis}_{\text{NADW}}} - r_{\text{C:N}} \cdot \text{NO}_{3_{\text{pre}_{\text{NADW}}}}^-)]
\end{aligned} \tag{11}$$

which can be generalized for any number $n$ of ventilation regions $i$ as:

$$\text{DIC}_{\text{dis}_{\text{global}}} + \text{DIC}_{\text{soft}_{\text{global}}} \approx r_{\text{C:N}} \cdot (\text{NO}_{3_{\text{global}}}^- + \text{NO}_{3_{\text{den,global}}}^-) + \sum_{i=1}^{n} f_i \cdot (\text{DIC}_{\text{dis}_i} - r_{\text{C:N}} \cdot \text{NO}_{3_{\text{pre}_i}}^-) \tag{12}$$

Thus, total carbon storage as soft and disequilibrium carbon (i.e. everything other than $\text{DIC}_{\text{sat}}$ and $\text{DIC}_{\text{carb}}$) varies with the global nitrate inventory, corrected for accumulated $\text{NO}_3^-$ loss to denitrification, and the difference between $\text{DIC}_{\text{dis}}$ and $r_{\text{C:N}} \cdot \text{NO}_{3_{\text{pre}}}^-$ in the polar oceans, modulated by their respective volume fractions. Although this nitrogen-based framework avoids the problem of C:P variability, it is not clear how large the effects of variable C:N might be in the real world. This could be a worthy topic for future exploration.

# 4 Conclusions

The conceptualization of ocean carbon storage as the sum of the saturation, soft tissue, carbonate, and disequilibrium components can greatly assist in enhancing mechanistic understanding (Williams and Follows, 2011; Ito and Follows, 2013; Ödalen et al., 2018). Our simulations indicate that the disequilibrium component may play a very important role, which has not been broadly appreciated. Changes in the physical climate states, as simulated by our model, tend to drive the soft tissue and the disequilibrium components in opposite directions. However, this is not necessarily true in the real ocean, given that the simulated anti-correlation is not mechanistically required, but instead arises from the fact that $f_{\mathrm{AABW}}$ and age $\times$ export are anti-correlated in the simulations. On the contrary, the radiocarbon analysis of Galbraith and de Lavergne (2018) suggests that the glacial ocean age was significantly greater than in the corresponding simulations, implying that this anti-correlation does not hold in reality. Our iron fertilization experiments explore another aspect of this decoupling, in which age $\times$ export increases despite no change in $f_{\mathrm{AABW}}$. There is plenty of scope for these to have varied in additional ways in the real world, not captured by our simulations, including the idealized mechanisms explored by Ödalen et al. (2018). Although the anti-correlation of $DIC_{\mathrm{dis}}$ and $DIC_{\mathrm{soft}}$ in our simulations results in small overall changes, their magnitudes are sufficient that their total scope for change exceeds that required to explain the glacial/interglacial $CO_2$ change.

Our results also show a surprising capacity for $O_2$ disequilibrium to develop in a cold state. We suggest that this reflects a high sensitivity of $O_2$ to sea ice when sea ice coverage reaches very high fractions. This generally unrecognized potential for sea ice coverage to cause large oxygen undersaturation may have contributed to very low $O_2$ in the Southern Ocean during glacial periods, as suggested by foraminiferal I/Ca measurements (Lu et al., 2015).

The results presented here suggest that disequilibrium carbon should be considered as a major component of ocean carbon storage, linked to ocean circulation and biological export in non-linear and interdependent ways. Despite these nonlinearities, the simulations suggest that the resulting global carbon storage can be well-approximated by simple relationships. We propose one such relationship, including the global nitrate inventory, and the $DIC_{\mathrm{dis}}$ and preformed $NO_3^-$ in ocean ventilation regions (eq. 12). As this study represents the result of a single model, prone to bias, it would be very useful to test our results using other GCMs including additional biogeochemical complexity. It would also be useful to consider how disequilibrium carbon can change under future $p$CO$_2$ levels, including developing observational constraints on its past and present magnitude, and exploring the degree to which inter-model variations in $DIC_{\mathrm{dis}}$ may contribute to uncertainty in climate projections.

*Code availability.* All model runscripts, code and simulation output are freely available from the authors, or can be obtained by download from https://earthsystemdynamics.org/cm2mc-simulation-library.

## Appendix A: DIC decomposition

DIC is treated as the sum of four components:

$$\text{DIC} = \text{DIC}_{\text{sat}} + \text{DIC}_{\text{dis}} + \text{DIC}_{\text{soft}} + \text{DIC}_{\text{carb}} \tag{A1}$$

$\text{DIC}_{\text{sat}}$ is the DIC at equilibrium with the atmosphere given the surface ocean temperature, salinity, and alkalinity and the atmospheric $p\text{CO}_2$ calculated following Zeebe and Wolf-Gladrow (2001):

$$\text{DIC}_{\text{sat}} = f(\text{T}, \text{S}, \text{alk}_{\text{pre}}, p\text{CO}_2) \tag{A2}$$

In this model, $\text{DIC}_{\text{soft}}$ is proportional to the utilized $\text{O}_2$, which is defined as the difference between preformed and total $\text{O}_2$, where the ratio of remineralized C to utilized $\text{O}_2$ ($r_{\text{C:O}_2}$) is 106:150.

$$\text{DIC}_{\text{soft}} = r_{\text{C:O}_2} \cdot (\text{O}_{2_{\text{pre}}} - \text{O}_2) \tag{A3}$$

DIC derived from $\text{CaCO}_3$ dissolution is proportional to the change in alkalinity, correcting for the additional change in alkalinity due to hydrogen ion addition during organic matter remineralization.

$$\text{DIC}_{\text{carb}} = 0.5 \cdot [(\text{alk} - \text{alk}_{\text{pre}}) + r_{\text{N:O}_2} \cdot (\text{O}_{2_{\text{pre}}} - \text{O}_2)] \tag{A4}$$

Preformed alkalinity, defined as the total alkalinity at the surface and treated as a conservative tracer, is calculated within the model framework but was not written out during the model runs. Therefore, we have reconstructed this parameter a posteriori for each model year through multilinear regressions as a function of century-averaged salinity (S), temperature (T), and preformed $\text{O}_2$, $\text{NO}_3^-$ and $\text{PO}_4^{3-}$, following the approach of Bernardello et al. (2014).

$$
\begin{aligned}
\text{alk}_{\text{pre}} = &(a_0 + a_1 \cdot \text{S}' + a_2 \cdot \text{T}' + a_3 \cdot \text{O}_{2_{\text{pre}}} + a_4 \cdot \text{NO}_{3_{\text{pre}}}^- + a_5 \cdot \text{PO}_{4_{\text{pre}}}^{3-}) \cdot \text{NAtl} \\
&+ (b_0 + b_1 \cdot \text{S}' + b_2 \cdot \text{T}' + b_3 \cdot \text{O}_{2_{\text{pre}}} + b_4 \cdot \text{NO}_{3_{\text{pre}}}^- + b_5 \cdot \text{PO}_{4_{\text{pre}}}^{3-}) \cdot \text{SO} \\
&+ (c_0 + c_1 \cdot \text{S}' + c_2 \cdot \text{T}' + c_3 \cdot \text{O}_{2_{\text{pre}}} + c_4 \cdot \text{NO}_{3_{\text{pre}}}^- + c_5 \cdot \text{PO}_{4_{\text{pre}}}^{3-}) \cdot (1 - \text{SO} - \text{NAtl})
\end{aligned} \tag{A5}
$$

where $\text{S}' = \text{S} - 35$, $\text{T}' = \text{T} - 20^\circ\text{C}$, the $a_i$ are determined by a regression in the surface North Atlantic, the $b_i$ for the SO, and the $c_i$ using the model output elsewhere in the surface. The tracers SO and NAtl are set to 1 in the surface Southern Ocean (south of 30°S) and the North Atlantic (north of 30°N), respectively, and are conservatively mixed into the ocean interior. This parametrization induces an uncertainty on the order of 1 $\mu$mol/kg in globally averaged $\text{DIC}_{\text{dis}}$ (see fig. A1). As discussed above, however, this is small compared to the signal seen over all simulations.

Finally, $\text{DIC}_{\text{dis}}$ has been back-calculated from the model output as a residual.

*Author contributions.* E. D. Galbraith conducted the model simulations, S. Eggleston performed the analysis, and both contributed to writing the manuscript.

*Competing interests.* The authors declare that they have no conflict of interest.

*Acknowledgements.* The authors would like to thank R. Bernardello for very helpful discussion. S. Eggleston was funded by a fellowship from the Swiss National Science Foundation. E. D. Galbraith acknowledges computing support from the Canadian Foundation for Innovation and Compute Canada, and financial support from the Spanish Ministry of Economy and Competitiveness, through the María de Maeztu
Programme for Centres/Units of Excellence in R&D (MDM-2015-0552).

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

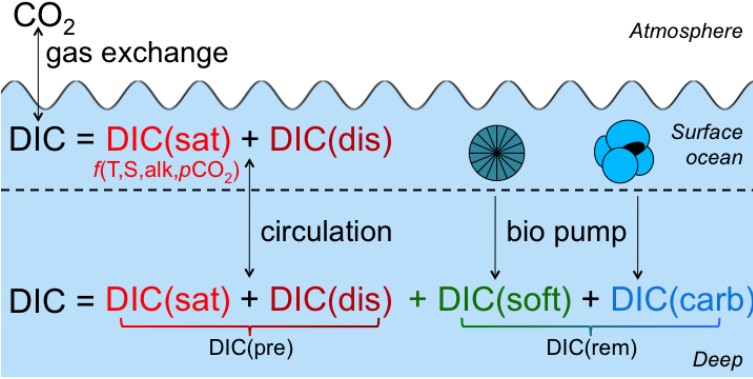

**Figure 1.** Illustration of the decomposition framework used for DIC in this paper. In the surface ocean, DIC is equal to $DIC_{pre} = DIC_{sat} + DIC_{dis}$. Carbon taken up by biology in the surface ocean sinks and remineralizes in the water column to add two additional components at depths: $DIC_{rem} = DIC_{soft} + DIC_{carb}$.

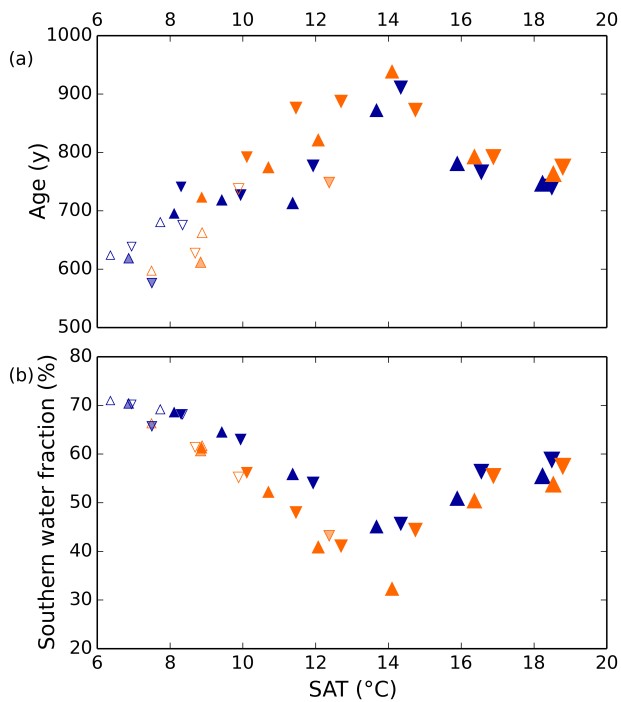

**Figure 2.** Global ocean ventilation and proportion of southern-sourced deepwater. (a) The global ocean average ideal age (low age corresponds to high ventilation) and (b) the fraction of water in the global ocean originating from the surface south of $30^{o}$S are anti-correlated over the range of SAT in these model runs. Orange and blue symbols represent high and low obliquity scenarios, respectively; triangles pointing upward and downward represent greater northern and southern hemisphere seasonality or precession $270^{o}$ and $90^{o}$, respectively; outlines are scenarios with LGM ice sheets; light shading indicates scenarios with LGM ice sheet topography but PI albedo. The size of the symbols corresponds to the SAT.

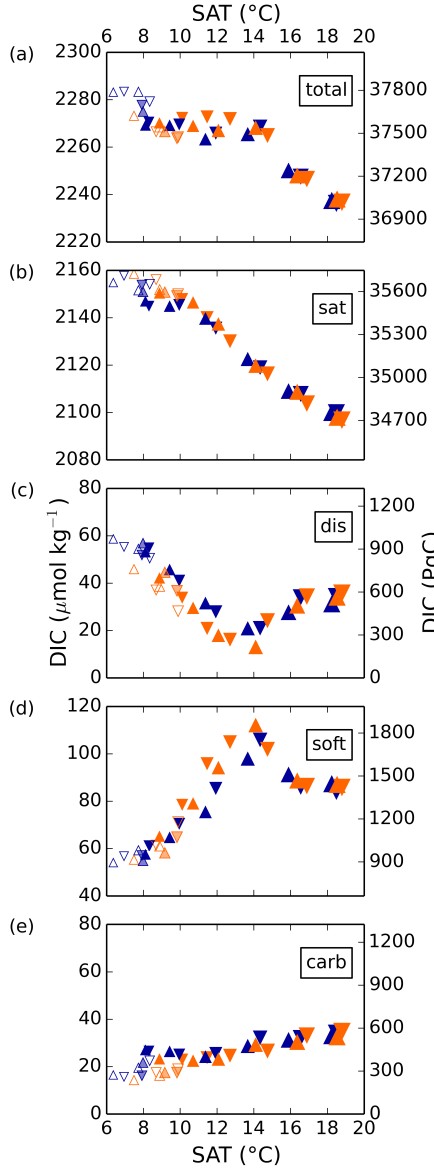

**Figure 3.** Global average DIC and separate components in simulations 1-36 as a function of surface air temperature. Orange and blue symbols represent high and low obliquity scenarios, respectively; triangles pointing upward and downward represent greater northern and southern hemisphere seasonality or precession 270° and 90°, respectively; outlines are scenarios with LGM ice sheets; light shading indicates scenarios with LGM ice sheet topography but PI albedo.

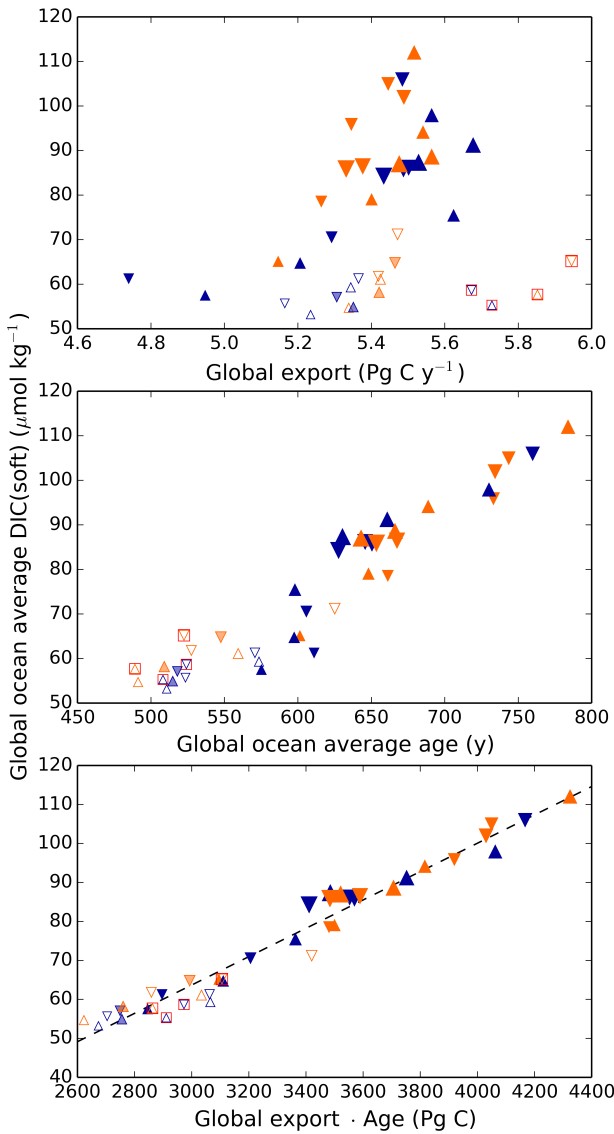

**Figure 4.** Globally averaged DIC$_{soft}$, or remineralized carbon from the soft tissue pump, can be approximated remarkably well by the global export flux of organic carbon at 100 m multiplied by the average age of the ocean. The latter is an ideal age tracer in the model that is set to 0 at the surface and ages by 1 y each model year in the ocean interior. Orange and blue symbols represent high and low obliquity scenarios, respectively; triangles pointing upward and downward represent greater northern and southern hemisphere seasonality or precession 270$^o$ and 90$^o$, respectively; outlines are scenarios with LGM ice sheets; light shading indicates scenarios with LGM ice sheet topography but PI albedo. Red boxes indicate Fe fertlization simulations (runs 37 – 40). The size of the symbols corresponds to the SAT.

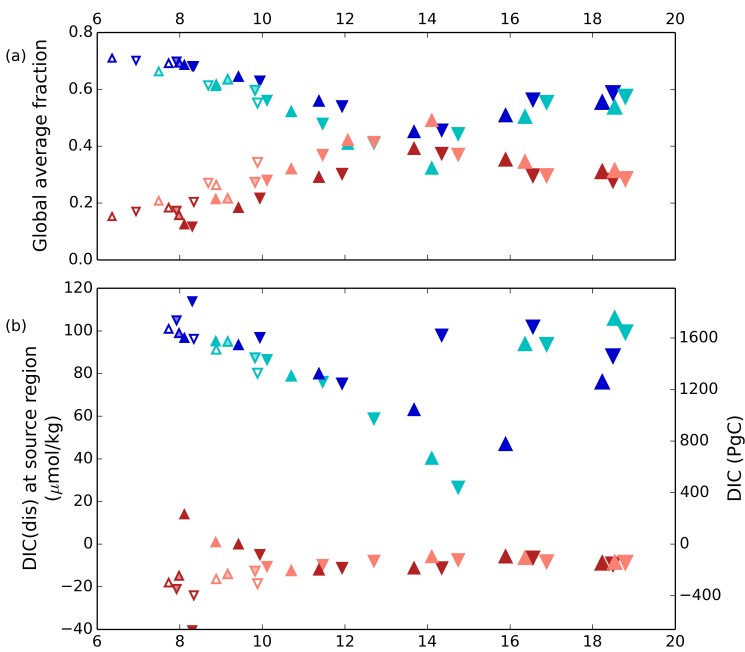

**Figure 5.** (a) Global average fraction of northern- (reddish colors) and southern-sourced (bluish colors) water. (b) Annual average values of $DIC_{dis}$ of these water masses determined at 25 m depth in the model during model years and at locations where deep convection occurs. Pink and cyan (red and blue) symbols represent high (low) obliquity scenarios; triangles pointing upward and downward represent greater northern and southern hemisphere seasonality or precession 270° and 90°, respectively; outlines are scenarios with LGM ice sheets; light shading indicates scenarios with LGM ice sheet topography but PI albedo. The size of the symbols corresponds to the SAT.

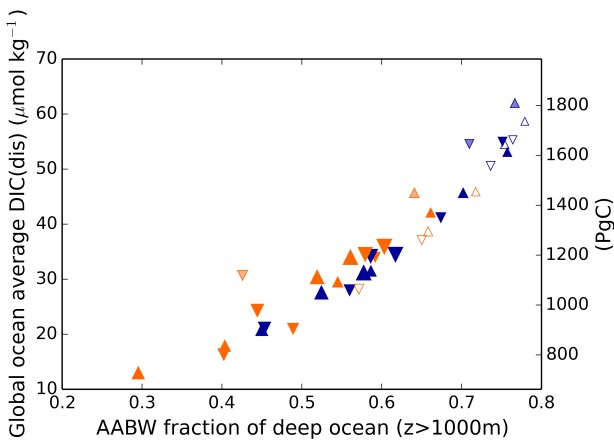

**Figure 6.** Global average DIC$_{dis}$ as a function of the fraction of the ocean below 1 km derived from the surface Southern Ocean. Orange and blue symbols represent high and low obliquity scenarios, respectively; triangles pointing upward and downward represent greater northern and southern hemisphere seasonality or precession 270$^o$ and 90$^o$, respectively; outlines are scenarios with LGM ice sheets; light shading indicates scenarios with LGM ice sheet topography but PI albedo. The size of the symbols corresponds to the SAT.

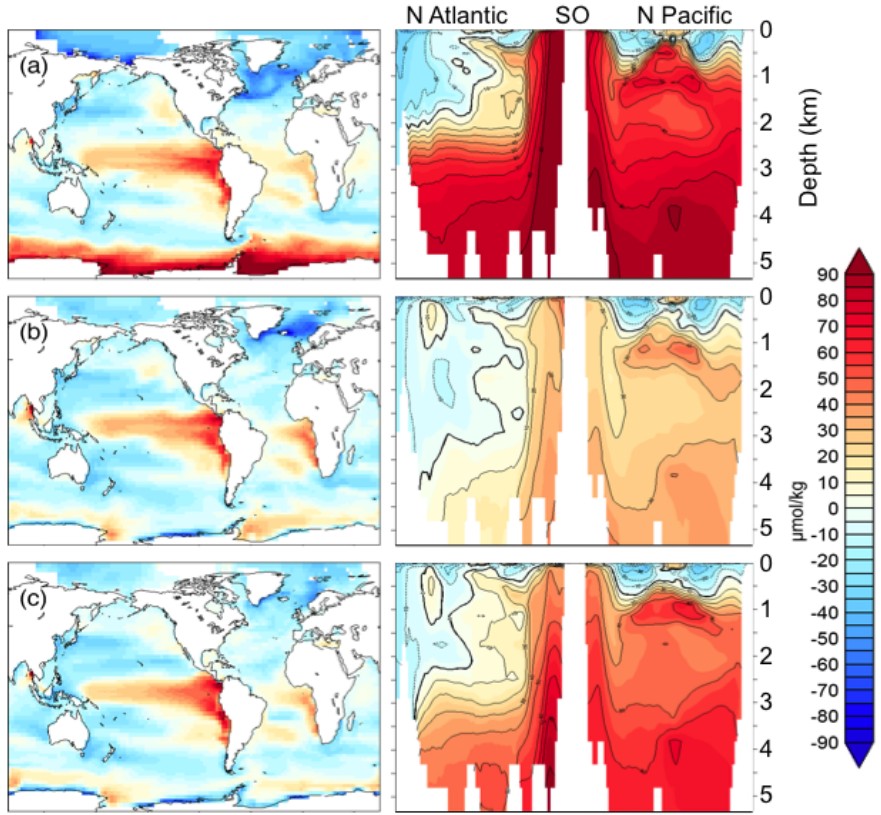

**Figure 7.** DIC$_{dis}$ ($\mu$mol kg$^{-1}$) for simulations (a) CW; (b) MW; (c) HW (see section 2.2). Depth transects represent the North Atlantic (left) - Southern Ocean (center) - North Pacific (right).

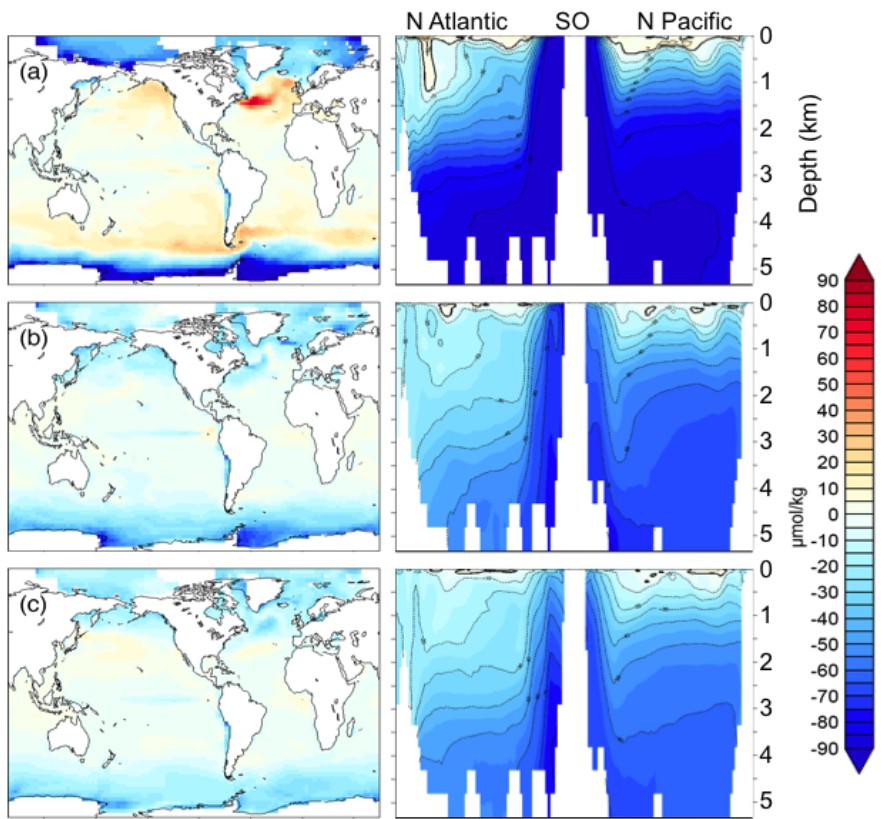

**Figure 8.** $O_{2_{dis}}$ ($\mu$mol kg$^{-1}$) for simulations (a) CW; (b) MW; (c) HW (see section 2.2). Depth transects represent the North Atlantic (left) - Southern Ocean (center) - North Pacific (right).

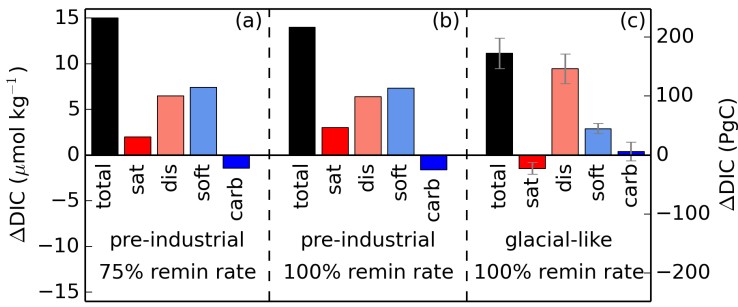

**Figure 9.** Iron-fertilized changes in total global average DIC and each of the components (iron fertilization simulation minus associated control run). Simulations were either run under pre-industrial or glacial-like conditions (in the case of the latter, results represent the average of the four GL runs, with error bars showing the standard deviation within the four runs), as well as 100% and 75% of the default remineralization rate of organic matter. The close agreement of the left and middle panels indicates that the effects of iron fertilization and changes in the remineralization rate are approximately linearly additive in this model.

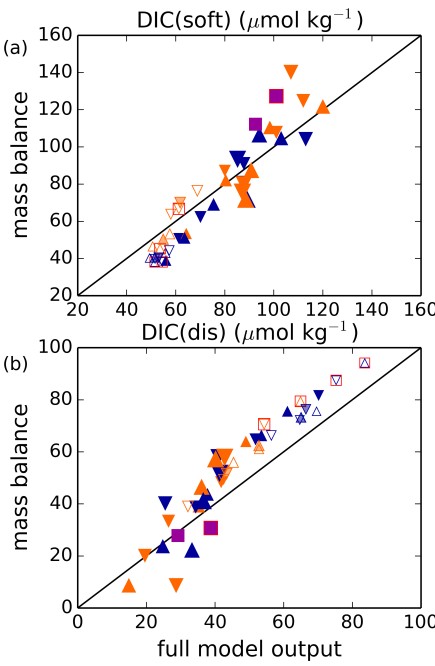

**Figure 10.** Simple parametrization of global DIC$_{dis}$ and DIC$_{soft}$ from water mass characteristics. Following eq. 12 for (a) DIC$_{soft}$ and (b) DIC$_{dis}$ separately, this model shows that the sum of the global average DIC$_{soft}$ and DIC$_{dis}$ at steady-state can be estimated fairly robustly as the result of a simple mass balance of the relevant parameters in the most important ventilating water masses. Here, we take into account upper-ocean water masses (above 1 km) formed in the North Pacific, North Atlantic, and Southern Ocean, and deep water masses formed in the North Atlantic and Southern Ocean. In each plot, the full model output is shown on the $x$-axis and the result of the mass balance approximation on the $y$-axis. Orange and blue symbols represent high and low obliquity scenarios, respectively; triangles pointing upward and downward represent greater northern and southern hemisphere seasonality or precession 270$^o$ and 90$^o$, respectively; outlines are scenarios with LGM ice sheets; light shading indicates scenarios with LGM ice sheet topography but PI albedo. The size of the symbols corresponds to the SAT. The purple square represents the pre-industrial simulation (run 41), and red boxes indicate Fe fertlization simulations (runs 37 − 40).

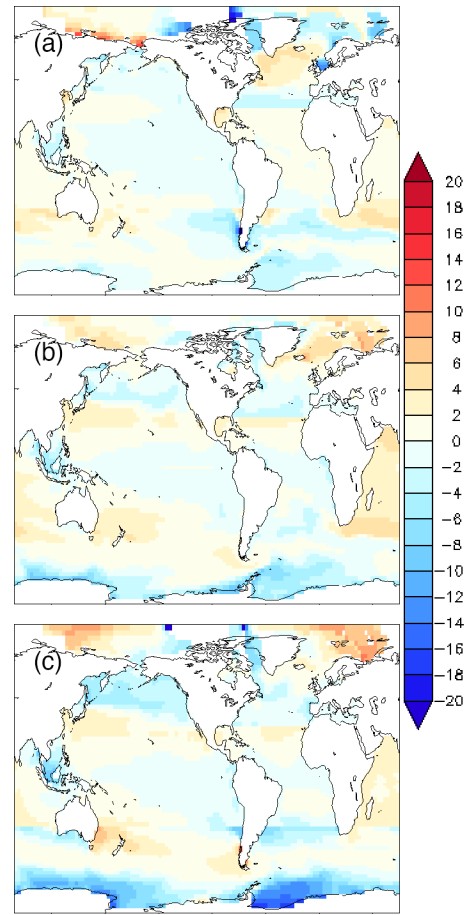

**Figure A1.** Shown is the difference between the exact $DIC_{dis}$ surface field in $\mu$mol kg$^{-1}$, where $DIC_{sat}$ has been calculated using the surface alkalinity ($alk[z=0] = alk_{pre}[z=0]$) and $DIC_{soft}[z=0] = DIC_{carb}[z=0] = 0$. Differences are shown for (a) LGM; (b) PI; (c) WF.

**Table 1.** Simulation overview. A total of 44 simulations were analyzed with varying radiative forcing (RF), obliquity, precession, ice sheets (PI = pre-industrial; LGM = Last Glacial Maximum reconstruction; LGM* = topography of LGM ice sheets but with PI albedo), and with and without iron fertilization. Runs 1 – 40 are described by Galbraith and de Lavergne (2018). Runs 43 and 44 and identical to 41 and 42 but the remineralization rate of sinking organic matter is reduced by 25%.

| run | RF in $p$CO$_2$ equivalents (ppm) | obliquity | precession | IS | Fe | remin |
|---|---|---|---|---|---|---|
| 1-24 | 180, 220, 270, 405, 607, 911 | 22º, 24.5º | 90º, 270º | PI | | |
| 25-28 | 220 | 22º, 24.5º | 90º, 270º | LGM | | |
| 29-32 | 180 | 22º, 24.5º | 90º, 270º | LGM* | | |
| 33-36 | 180 | 22º, 24.5º | 90º, 270º | LGM | | |
| 37-40 | 180 | 22º, 24.5º | 90º, 270º | LGM | X | |
| 41 | 270 | 23.4º | 102.9º | PI | | |
| 42 | 270 | 23.4º | 102.9º | PI | X | |
| 43 | 270 | 23.4º | 102.9º | PI | | 75% |
| 44 | 270 | 23.4º | 102.9º | PI | X | 75% |