# Peer review of "The devil's in the disequilibrium: multi-component analysis of dissolved carbon and oxygen changes under a broad range of forcings in a general circulation model"

_Biogeosciences, 2017_

## Referee Comment (RC1) · Anonymous Referee #1 · 16 Oct 2017

This paper describes some 44 simulation scenarios with the GFDL climate model in order to understand which processes might be responsible for the glacial $CO_2$ drawdown, which is observed in the ice core data.

The paper makes an separation of DIC into the soft issue pump, the carbonate pump, and saturation and disequilibrium, and might in principle be worth publishing. However, the form of presentation needs some fundamental rework for various reasons, which are find below. I find both the representation of the text and of the results in the figure very sloppy and full of not very detailed descriptions.

[Figure]

Main issue: As already indicated by the title of the paper the disequilibrium component of DIC is the part of the carbon fluxes which seemed to be of major relevance, but which seemed to have been neglected in previous papers. Disequilibrium DIC ist the difference between saturated DIC (surface ocean DIC in equilibrium with the atmosphere, here defined by constant 270 ppm) and actual DIC. Marine carbon uptake is slowed down by the marine carbonate chemistry, because DIC for present surface ocean conditions is found as 1% CO2, 90% HCO3 and and 9% CO3, but only the 1% CO2 can exchange with the atmosphere. My understanding of the disequilibrium DIC is therefore, that it represents a different way of saying, that oceanic carbon uptake is restricted by marine carbon chemistry. For example, a change in atm CO2 by 10% leads only to a change in DIC of about 1%. This effect of the chemistry is summarized by the Revelle or buffer factor $R = (\Delta CO2/CO2)/(\Delta DIC/DIC)$ which is around 10, but various between 8 and 15 (e.g. Sabine et al., 2004, Science). Since full carbon cycle models all include the relevant carbonate chemistry, this effect is always included, and I am missing a connection of the newly analysed disequilibrium component of DIC with this issues. Maybe the disequilibrium DIC is not that new at all.

Therefore, I have the feeling, the paper is lost in details, but misses more strength on a red line. I also believe the scenario definition with fixed and constant CO2 at 270 ppm is a major drawback in the value of the paper, since it implies that all quantifications of the fluxes have to be treaded very carefully: They have to be wrong, since the gas exchange of CO2 heavily depends on the pressure gradient in CO2 between atmosphere and ocean.

Other majors:

- The authors have chosen to keep atmospheric CO2 fixed, so they calculate changes in oceanic DIC only as a function of prescribed CO2 (always 270 ppm), which ignores dynamic aspects of the gas exchange, that largely depend on the surface ocean-atmospheric difference in pCO2. This is a significant simplication,

which reduces the significance of the quantification of the process separation a lot. It implies, that atmospheric CO2 concentration is not a dynamic part of the carbon cycle analysis anymore. CO2 is nevertheless varied, but only to generate different background climatologies, implying only the radiative forcing of CO2 is used here. I therefind find the description of all scenarios and results highly confusing, they should not be defined by the prescribed atm CO2 value, because this is not considered in the carbon cycle change, but by the resulting global annual mean surface temperature changes, $\Delta T$. I therefore expect, that the authors, (i) calculate $\Delta T$, probably with respect to their control simulation (probably the one with CO2 = 270 ppm), and (ii) use $\Delta T$ when describing the scenarios, in the text, in Table 1, and in the Figures (e.g. x axis of Fig 1, 4). Since changes in obliquity, precession, and land ice sheet might also change $\Delta T$, they might also be more specific and call this $\Delta T_{CO2}$, but they might then also analyse the temperature change related to these other processes. If they then plot results in Fig 1 as function of Delta T, a lot of the various scenarios with similar CO2, but different other boundary conditions might then separte in $\Delta T$, and might be easier to be identified. Right now Fig 1 is an mess, with various symbols plotted on top of each other. If thss step does not improve figure 1, the authors might also consider to plot Fig 1 as bar charts, where different scenarios by definition are plotted NEXT and not ON TOP of each other.

• Anomalies on mean ocean DIC are analysed given in $\mu$mol/kg. However, I would find it much more helpful, if the amount of carbon taken up by the ocean would be given in terms of PgC (= GtC), which should be transferable easily (if the mean density of water and the volume of the ocean are known). Maybe, if the authors insist on their view on the system ($\mu$mol/kg), they might simply add a 2nd y-label (right-handside) with PgC. This would help a lot, since it is not clear to me, if the setup of the climate model in the LGM mode (more land ice) would also imply less ocean volume, which would directly affect the concentration of DIC in $\mu$mol/kg,

but not total amount ocean C in PgC. The discussion how much a change in DIC would change atm CO2 (Discussion, page 6) can be simplified a lot by stading the change in oceanic DIC in PgC.

- In the text various times a change in carbon due to a change in ocean circulation is seen (e.g. page 5, line 26, line 29, page 6, line 2), however, the ocean circulation state is never described in the draft. The reader does not know which of the scenarios has a slower ocean overturning. If this relationship should be kept in the text, some further details (ocean circulation analysis) is needed. My impression is, this might be found in the draft Galbraith + de Lauvergne (submitted), but this is not accessible. So, either the resubmission of this paper has to wait for the other paper, or some of these analysis need to be repeated here.

- I disagree with the potential usability of the quantification of the soft tissue pump on page 6, implying that the equations might be used as simplification in more complex models. First. eq 6 should be deleted, since the change in atm CO2 can only be quantified once atm CO2 is calculated dynamically. Second, the ocenic DIC uptake (eq 5) has also as major weakness to deal with the overall setup, that include constant and identical CO2=270 ppm, which implies, that quantification have to be discussed with care. Here, they are taken as given quantification of the soft tissue pump, which might even be used elsewhere. I do not think, this is the case for the reason given above. Furthermore, the sentence, page 7, line 22 "In contrast, the model suggests that greater ocean ventilation rates in the glacial state would have led to reduced global $DIC_{soft}$." and the following discussion is coming from nowhere and is not supported with any data. We know nothing on the ocean ventilation stage so far. Later-on, in section 4.7 it is argued, that interactive CO2 would only lead to minor effect. I strongly disagree, since the atmosphere-surface ocean gas exchange is a function of the pCO2 difference of both.

- Whenever a statement is made, which variable change how much, which is also find in a figure, please include a reference to this figure.

Minors (chronologically):

- The meaning of the four separations of DIC ($DIC_{sat}$, $DIC_{dis}$, $DIC_{soft}$, $DIC_{carb}$) are explained in the abstract, but not again in the introduction. I believe, this should be repeated around line 10 (page 2), to make the main text independent from the abstract.

- page 2, line 24: "$DIC_{soft}$ depends ... on the flushing rate of the deep ocean which clears out accumulated $DIC_{soft}$". This is a bit sloppy and not correct: $DIC_{soft}$ depends on the ocean circulation as a whole, the surface to deep ocean transport also transports it to the deep ocean.

- page 3, line 11: "preformed DIC" is not explained/defined her, onlxy later on page 4.

- page 3, line 25f: "Here, we use a fully-coupled general circulation model (GCM) to investigate the potential importance of $DIC_{dis}$ in altering air-sea CO2 partitioning on long timescales." This aim of the paper is not given as such in the abstract, but should be contained there.

- page 3, line 30: "prescribe a constant CO2": It is not only constant in individual experiments, it is also identical in all experiments as 270 ppm. This should be clarified here directly, because it implies, that the atm CO2 is only a driver of climate, but not of carbon cycle changes.

- page 4, line 5: "static land and ice sheet". Please explain, or should this read "static land ice sheets"?

- page 4, line 10: The paper Galbraith and de Lavergne (submitted) is missing in the reference list, but need to be included there, at best with a link to an accessible version. If not, maybe introduce this information only, once the paper is accessible somewhere, eg in the next iteration of the paper.

- page 4, experiments: As already given above, the different CO2 levels should be transfered into $\Delta T$. Also give a brief reasoning for your choices of CO2 here, e.g. why is the reference CO2 270 ppm, and not 278 ppm, as usual, why 607 and 911 ppm?

- page 4, experiments: Orbital parameters: Eccentricity seemed to have not changed, but nevertheless, state its values, and for which climate state it is typical. Also state, typical values of obliquity and precession for today and LGM.

- page 4, line 17: Iron fertilization: Did I get it right, that the glacial dust fields only change iron availability, but not the radiative forcing? Please clarify. Furthermore, the field in the cited paper (Nickelsen and Oschlies 2015) are taken from Mahowald et al 2006, which should be state here. Also be aware (and potentially discuss), that, at least to my knowledge, more recent LGM dust fields of the Mahowald groud differ to the dust field published in 2006 (especially in the high-latitudes), (e.g. Albani et al 2012 (Clim Dyn), Albani et al 2016 (GRL), which might have an impact on Southern Ocean iron fertilization.

- page 4, line 18: "standard deviation of only 4 $\mu$mol/kg". I do not see any error bars in Fig 1, and in DIC$_{carb}$ the values vary by 20 to 35 $\mu$mol/kg. Something is wrong here.

- page 4, line 23: "under nutrient depletion experiment". Which one would that be? I only see Fe fertilization expliements, which would be the opposite.

- page 5ff: I suggest to combine "3 Results" and "4 Discussions" into "3 Results and Discussions" since sec 3 so far is pretty short and sec 4 also includes results.

- page 6, line 30: Define "ideal age".

- page 6, line 31,32: "quantitative strength of the relationship is striking". It is, what I would expect. The interesing part for me would actually be, to see the individual contribution of the two variables combined in Figure 3, so please also plot "global export" and "age" and define both precisely (are they averaged, if yes, over which results?).

- page 7, eq 5,6: These regressions equations are of very limited use, since atm CO2 has been kept constant at 270 ppm, and the gas exchange is a linear function of the atmosphere-surface ocean CO2 difference.

- page 7, line 2": We know nothing on ocean ventilations rates yet.

- page 8, line 2: Here surface and deep ocean are split at 1 km depth, but in Fig 5 at 500 m depth, Please be consistent.

- page 8, Eq 7: Here, the Southern Ocean contributen is termed $f_{AABW}$, while in Eq 10 it is termed $f_{SO}$. Please be consistent.

- page 8, line 15: "hot and cold climate state": Be more precise what this means in terms of scenarios. This sentence probably refers to Fig 4, but no reference to it was given.

- page 8: I suggest to combine the to subsection 4.2 and 4.3 to one subsection.

- page 8, line 26: "when deep convection is occuring" When does it occur and where, scenario?

- page 8, line 27: "... ventilation rates are high at both the cold and hot extremes". Show plot.

[Figure]

- page 9, line 9: $O_{2,dis}$ is not defined (I believe). You might add why O2 equilibrates an order of magnitude faster than CO2 (no bottleneck of the carbonate chemistry during oceanic uptake).

- page 9, line 11: "$O_{2,dis}$ in the SO is as high as 100 $\mu$mol/kg". I believe this is wrong, should be "as low as -100 $\mu$mol/kg", and the reference to the figure is needed (Fig 9). $O_{2,dis}$ seems to be largely anticorrelated to DIC$_{dis}$, but this was never mentioned as such.

- page 9, line 21: "reduced sensitivity of export" to what?

- page 9, last line: I do not undertstand why the two effects of iron fertilization and remineralization rate should be linearly additive.

- page 10ff: Change $NO_3$ in $NO_3^-$

- page 10, line 13: What happens to you framework, if N:C ratios are not constant, as for example in postulated by Geider et al. (1998)?

  Geider, R. J.; MacIntyre, H. L. Kana, T. M. A dynamic regulatory model of phytoplanktonic acclimation to light, nutrients, and temperature, Limnology and Oceanography, 1998, 43, 679-694.

- page 10ff: Unified framework: Here it is argued that the NO$_{3,pre}$ can be split in the contributions from Southern Ocean (SO), North Atlantic (NAtl) and North Pacific (NPac), later-on the argument is made that the North Pacific can be neglected due to missing deep convection. The contributions to DIC$_{dis,deep}$ early in the draft (sec 4.2) was only split between NAtl and SO, but not NPac. Please be consistent in both approaches.

- page 11, Eq 14 and 15: I have the feeling the factors $V_i/V_{total}$ which are included in Eq 14 are missing in Eq 15, but should still be included here.

- page 11, line 17: RMSE: How has this been found? Is this the mean difference to the 1:1 line in Fig 10?

- page 11, Section 4.7: I think this section might be called "General discussion". I do not think the naming of "$DIC_{dis}$ nadir" is helpful here, pleasxe consider other wording.

- page 12, lines 7-13: This paragraph is highly speculative and with the given support not justified. We know nothing on AABW formation for different climate state and the fixed, constant atmospheric CO2 boundary condition hinders in my view that such bold hypothesis are made based on the made analysis.

- page 13, line 13: "ratio of remineralized to UTILIZED O2"

- page 13, line 18ff: Preformed alkalinity ($alk_{pre}$) is defined as total alkalinity at the surface. However, $alk_{pre}$ is then calculated from T, S, NO3, PO4. Does this imply, total alkalinity is not followed as an independent tracer in the carbon cycle of the ocean in the model? If not, this need some explanation, since normally the full carbon cycle needs 2 variables to be fully prognostic, typically the conservative tracers DIC and ALK are taken for that (from which pH, CO2, CO3, HCO3 are then calculated. Please clarify, and explain.

- page 14: Eq A6 is trivial, its another version of Eq A1.

- Figures: Make plots larger to have larger font size, include sub-plot names (eg Fig 1a,bc), check if units are always given (missing in Fig 11). In all figures each caption needs to have the full explanation of what is seen here (and not refering to Fig 1 as done so far).

- Fig 1: Plot all 5 subfigures on top of each other, planing, that this will fill 1 column in the final layout text, change x axis to $\Delta T$, consider plotting it as bar charts,

clarify what you mean in the caption with changed seaonality (probably a change in pressession).

- Fig 2: If results are averages from 4 runs (as said here) give mean and error from averaging.

- Fig 3: X axis: "Global export times age". "Global export" here implies only export of organic C or also CaCO3 export?

- Fig 4: Why do you take DIC$_{dis}$ at 100m water, and not the average over the mixed layer?

- Fig 6,8,9,11: Define scenarios "glacial" and "interglacial". Are these averages over several scenarios? Unit is missing in Fig 6.

- Fig 7: I find this figure, highly confusing (not clear what lines represent), and not necessary at all.

- Fig 10: Caption: Please reduce text and refer to where the Eq is found in text, e.g. lower plot y axis follows Eq 15. I am not sure for the other 2 plots there is a complete Eq contained in the theoretical framework, if not, please extend.

- Fig 11: I am not sure, this Fig is necessary.

---

## Referee Comment (RC2) · Anonymous Referee #2 · 14 Nov 2017

This study deals with an interesting aspect of the global carbon cycle, relevant to understanding past natural changes of atmospheric CO2 mixing ratio reconstructed from geological archives. As such it is relevant to the readership of Biogeosciences.

The decomposition of Dissolved Inorganic Carbon (DIC) into component parts based on a process-driven approach is an established procedure in the literature. This main advance here is a detailed look at a particular component of DIC (the 'disequilibrium' DIC) that has been overlooked in some previous studies.

[Figure]

Generally, the literature is well covered by the references. Although, I believe DeVries et al. "The sequestration efficiency of the biological pump", GRL (2012) is directly relevant to Figure 3 in this study, and should be considered when the findings linked to figure 3 are discussed.

The main findings are the de-coupling of the DIC_dis from the (previously) expected behaviour based on DIC_soft in some circumstances. This is shown in a number of idealised numerical model experiments. This finding is important and relevant to the literature.

Specifically, equation (15), figure (10) and the insight gained from these for the idealised simulations are important and deserve to be published provided they are found to be robust.

With the manuscript in its present form, I have some reservations as to whether the findings are robustly supported by the work undertaken. I am unable to say for certain if this can be addressed by re-writing of the manuscript, or would require altered or additional numerical model runs. Below I detail my concerns on this issue.

Major concerns:

(1) Understanding the methods described in the text and their relation to the figures.

As noted by Anonymous Reviewer 1, this study uses a fixed and constant atmospheric $CO_2$ mixing ratio of 270 ppm in the numerical model experiments. While I see nothing intrinsically wrong with this approach, the manuscript as it is does not adequately describe the impacts of this choice on their results.

For example, consider Figure 1 and accompanying text: On p5 lines 15-17, the manuscript states how a constant $CO_2$ for gas exchange of 270 ppm is used. Then, the manuscript states how the largest changes in DIC_sat is driven by the changes in $\ln(CO_2)$. Figure 1 does indeed plot DIC_sat against $CO_2$ with $CO_2$ shown to vary.

I do not understand this figure or the text: precisely how is DIC_sat a function of

**BGD**

$\ln(CO_2)$ if a constant $CO_2$ of 270 ppm is used for gas exchange. If a constant $CO_2$ of 270 ppm is used for gas exchange in the numerical simulations, then surely the $CO_2$ cannot be changing on the axes in figure 1. Also, $DIC_{sat}$ should always be calculated relative to a $CO_2$ of 270 ppm, and so $DIC_{sat}$ will not change with $\ln(CO2)$.

The issue recurs in Figure 2, the caption to which indicates that some experiments are run at 180 ppm $CO_2$ concentration.

This confusion is critical for assessment of the manuscript in its current form (e.g. also see major point 2, which derives from this).

Note that there are other studies in this topic that have dealt with similar issues well. For example, Marinov et al (2008) (cited by this manuscript) uses GCM simulations with a fixed temperature for air-sea gas exchange, but a varying dynamical temperature for ocean circulation. In that manuscript, the issue is well described and the findings are clear.

Is the issue here that one $CO_2$ is used for the radiative forcing of climate, but another $CO_2$ is used for air-sea gas exchange?

I cannot tell precisely with the manuscript in its present form. If this is the case, then all mentions of $CO_2$ in ppm {and $\ln(CO2)$} that are different from 270ppm could be converted to radiative forcing (in W/m^2) with respect to $CO_2$ = 270 ppm. For example, $CO_2$ = 180ppm would be re-defined as 'Glacial radiative forcing' (or a numerical radiative forcing of $\sim$ -2.2 W/m^2).

For example: P11 Line 20 to line 24 reads: "The model simulations show a clear minimum $DIC_{dis}$ at intermediate $CO_2$ (270-405 ppm). . . . the $CO_2$ driving gas exchange . . . is held fixed at 270 ppm."

This could be changed to something like: "The model simulations show a clear minimum $DIC_{dis}$ at intermediate Radiative Forcing (0 – 2.2 Wm-2). . . . the $CO_2$ driving gas exchange . . . is held fixed at 270 ppm."

I am currently confused by the way this issue is written about in the study. Please clarify.

(2) DIC_sat and DIC_dis definitions:

There are two ways of defining DIC_sat and DIC_dis in the present-day system of rising $CO_2$, or over different periods when $CO_2$ has changed in the past.

Firstly, you can define both DIC_sat and DIC_dis relative to some fixed atmospheric $CO_2$ concentration (such as the preindustrial). Or secondly, you can define them relative to the current atmospheric $CO_2$ concentration at a particular point in time (for example it would be relative to 180 ppm at the LGM, or ∼400ppm in the present day).

This choice makes a big difference. Consider the present day: if DIC_sat is defined relative to present day atmospheric $CO_2$ then DIC_dis is small in the surface ocean and negative in the deep ocean. However, if DIC_sat is defined relative to preindustrial $CO_2$ then DIC_dis is positive in the surface ocean and ∼ zero at depth.

In the original discussions of DIC_sat (p2, lines 14-17) and DIC_dis (p3, lines 14-34), it is unclear whether a fixed or rising $CO_2$ concentration will be used to define DIC_dis and DIC_sat.

It eventually becomes clear (I think) that in this study, the DIC_sat is calculated relative to a fixed $CO_2$ of 270 ppm (page 4, line 15). However, the point is only made when discussing the numerical model set up.

A clearer indication of how DIC_sat and DIC_dis are treated in this study from the outset is required. Especially given the confusing 'fixed but changing' $CO_2$ issue from my other major concern. If the experiments are run with $CO_2$=180ppm, and DIC_sat is defined relative to 270ppm then this will have a large impact on the results.

---

## Author Comment (AC2) · 19 Dec 2017

The devil's in the disequilibrium: sensitivity of ocean carbon storage to climate state and iron fertilization in a general circulation model
Sarah Eggleston and Eric D. Galbraith
bg-2017-328

Response: Reviewer 2

This study deals with an interesting aspect of the global carbon cycle, relevant to un- derstanding past natural changes of atmospheric CO2 mixing ratio reconstructed from geological archives. As such it is relevant to the readership of Biogeosciences.

The decomposition of Dissolved Inorganic Carbon (DIC) into component parts based on a process-driven approach is an established procedure in the literature. This main advance here is a detailed look at a particular component of DIC (the 'disequilibrium' DIC) that has been overlooked in some previous studies.

Generally, the literature is well covered by the references. Although, I believe DeVries et al. "The sequestration efficiency of the biological pump", GRL (2012) is directly relevant to Figure 3 in this study, and should be considered when the findings linked to figure 3 are discussed.

**We agree this is a relevant references and will add it in the revision.**

The main findings are the de-coupling of the DIC_dis from the (previously) expected behaviour based on DIC_soft in some circumstances. This is shown in a number of idealised numerical model experiments. This finding is important and relevant to the literature.

**We thank the reviewer for their support.**

Specifically, equation (15), figure (10) and the insight gained from these for the ide- alised simulations are important and deserve to be published provided they are found to be robust.

With the manuscript in its present form, I have some reservations as to whether the findings are robustly supported by the work undertaken. I am unable to say for certain if this can be addressed by re-writing of the manuscript, or would require altered or additional numerical model runs. Below I detail my concerns on this issue.

1. Understanding the methods described in the text and their relation to the figures.
   As noted by Anonymous Reviewer 1, this study uses a fixed and constant atmospheric CO2 mixing ratio of 270 ppm in the numerical model experiments. While I see nothing intrinsically wrong with this approach, the manuscript as it is does not adequately describe the impacts of this choice on their results.
   For example, consider Figure 1 and accompanying text: On p5 lines 15-17, the manuscript states how a constant $CO_2$ for gas exchange of 270 ppm is used. Then, the manuscript states how the largest changes in DIC_sat is driven by the changes in $\ln(CO_2)$. Figure 1 does indeed plot DIC_sat against CO_2 with CO_2 shown to vary.
   I do not understand this figure or the text: precisely how is DIC_sat a function of $\ln(CO_2)$ if a constant CO_2 of 270 ppm is used for gas exchange. If a constant CO_2 of 270 ppm is used for gas exchange in the numerical simulations, then surely the CO_2

cannot be changing on the axes in figure 1. Also, DIC_sat should always be calculated relative to a $CO_2$ of 270 ppm, and so DIC_sat will not change with ln(CO2).

The issue recurs in Figure 2, the caption to which indicates that some experiments are run at 180 ppm $CO_2$ concentration.

This confusion is critical for assessment of the manuscript in its current form (e.g. also see major point 2, which derives from this).

Note that there are other studies in this topic that have dealt with similar issues well. For example, Marinov et al (2008) (cited by this manuscript) uses GCM simulations with a fixed temperature for air-sea gas exchange, but a varying dynamical temperature for ocean circulation. In that manuscript, the issue is well described and the findings are clear.

Is the issue here that one $CO_2$ is used for the radiative forcing of climate, but another $CO_2$ is used for air-sea gas exchange?

I cannot tell precisely with the manuscript in its present form. If this is the case, then all mentions of $CO_2$ in ppm [and ln(CO2)] that are different from 270ppm could be converted to radiative forcing (in W/m^2) with respect to $CO_2$ = 270 ppm. For example, $CO_2$ = 180ppm would be re-defined as 'Glacial radiative forcing' (or a numerical radiative forcing of _ -2.2 W/m^2).

For example: P11 Line 20 to line 24 reads: "The model simulations show a clear minimum DIC_dis at intermediate $CO_2$ (270-405 ppm). : : : the $CO_2$ driving gas exchange : : : is held fixed at 270 ppm."

This could be changed to something like: "The model simulations show a clear minimum DIC_dis at intermediate Radiative Forcing (0 – 2.2 Wm-2). : : : the $CO_2$ driving gas exchange : : : is held fixed at 270 ppm."

I am currently confused by the way this issue is written about in the study. Please clarify.

**Indeed, different $CO_2$ levels are used for the radiative forcing, which is varied from 180 to 911 ppm in these simulations, and $CO_2$ used for the biogeochemistry, including air-sea gas exchange, which is always 270 ppm. Given the similar confusion raised by Reviewer 1, we will refer only to the different climate forcings in terms of the mean surface temperature ($\Delta$T). We feel this will alleviate a great deal of misunderstanding.**

2. DIC_sat and DIC_dis definitions:
   There are two ways of defining DIC_sat and DIC_dis in the present-day system of rising $CO_2$, or over different periods when $CO_2$ has changed in the past.

   Firstly, you can define both DIC_sat and DIC_dis relative to some fixed atmospheric $CO_2$ concentration (such as the preindustrial). Or secondly, you can define them relative to the current atmospheric $CO_2$ concentration at a particular point in time (for example it would be relative to 180 ppm at the LGM, or _400ppm in the present day).

   This choice makes a big difference. Consider the present day: if DIC_sat is defined relative to present day atmospheric $CO_2$ then DIC_dis is small in the surface ocean and negative in the deep ocean. However, if DIC_sat is defined relative to preindustrial $CO_2$ then DIC_dis is positive in the surface ocean and _ zero at depth.

   In the original discussions of DIC_sat (p2, lines 14-17) and DIC_dis (p3, lines 14-34), it is unclear whether a fixed or rising $CO_2$ concentration will be used to define DIC_dis and DIC_sat.

   It eventually becomes clear (I think) that in this study, the DIC_sat is calculated relative to a fixed $CO_2$ of 270 ppm (page 4, line 15). However, the point is only made when discussing the numerical model set up.

A clearer indication of how DIC_sat and DIC_dis are treated in this study from the outset is required. Especially given the confusing 'fixed but changing' $CO_2$ issue from my other major concern. If the experiments are run with $CO_2$=180ppm, and DIC_sat is defined relative to 270ppm then this will have a large impact on the results.

**We agree this is potentially confusing. We will add text to the introduction to precisely define the terms, as we use them. We will also add a new figure to illustrate these important concepts clearly. In our usage, both DIC(sat) and DIC(dis) are determined only in the surface layer and are propagated into the interior by mixing and advection. Thus, under a transient change of $CO_2$, the surface values would evolve as the $CO_2$ changes, and the values propagated into the interior would follow.**

---

## Author Response (AR1)

The devil's in the disequilibrium: sensitivity of ocean carbon storage to climate state and iron fertilization in a general circulation model
Sarah Eggleston and Eric D. Galbraith
bg-2017-328

Response: Reviewer 1

This paper describes some 44 simulation scenarios with the GFDL climate model in order to understand which processes might be responsible for the glacial CO2 drawdown, which is observed in the ice core data.

The paper makes an separation of DIC into the soft issue pump, the carbonate pump, and saturation and disequilibrium, and might in principle be worth publishing. However, the form of presentation needs some fundamental rework for various reasons, which are find below. I find both the representation of the text and of the results in the figure very sloppy and full of not very detailed descriptions.

1. Main issue: As already indicated by the title of the paper the disequilibrium component of DIC is the part of the carbon fluxes which seemed to be of major relevance, but which seemed to have been neglected in previous papers. Disequilibrium DIC ist the difference between saturated DIC (surface ocean DIC in equilibrium with the atmosphere, here defined by constant 270 ppm) and actual DIC. Marine carbon uptake is slowed down by the marine carbonate chemistry, because DIC for present surface ocean conditions is found as 1% CO2, 90% HCO3 and and 9% CO3, but only the 1% CO2 can exchange with the atmosphere. My understanding of the disequilibrium DIC is therefore, that it represents a different way of saying, that oceanic carbon uptake is restricted by marine carbon chemistry. For example, a change in atm CO2 by 10% leads only to a change in DIC of about 1%. This effect of the chemistry is summarized by the Revelle or buffer factor $R = (\Delta CO2/CO2)/(\Delta DIC/DIC)$ which is around 10, but various between 8 and 15 (e.g. Sabine et al., 2004, Science). Since full carbon cycle models all include the relevant carbonate chemistry, this effect is always included, and I am missing a connection of the newly analysed disequilibrium component of DIC with this issues. Maybe the disequilibrium DIC is not that new at all.

**Thank you for the comment, which has indicated to us that the terms were not sufficiently described in the initial submission. We will elaborate on the discussion of DIC(dis) in the introduction of the text to clarify what it represents, and we plan to add a new figure to illustrate the concepts more clearly. In short, the disequilibrium component is indeed related to the carbonate chemistry, which slows the air-sea exchange as described, but it is also a function of biological uptake, ocean circulation, and gas exchange. It can be seen as the net result of all non-equilibrium processes on the DIC concentration within the surface layer.**
**It is also absolutely true that the disequilibrium effect is included in all full carbon cycle models (though, as pointed out on page 3 in lines 18-20, this component is by definition excluded from models when air-sea gas exchange is assumed to be infinitely fast). The purpose in recognizing it as an explicitly defined component of carbon storage is for understanding the underlying mechanisms. For example, in studying the glacial-interglacial change, the focus is generally on the importance of the saturation and soft tissue "pumps"**

**(e.g., Archer et al., 2000; Sigman and Boyle, 2000). We note that Ödalen et al. (Biogeosciences Discussions, 2017) use a similar approach, with a similar motivation.**

2. Therefore, I have the feeling, the paper is lost in details, but misses more strength on a red line.

**Thank you for the suggestion to highlight the take-home messages more clearly; we intend to make this a central goal of the revision.**

3. I also believe the scenario definition with fixed and constant CO2 at 270 ppm is a major drawback in the value of the paper, since it implies that all quantifications of the fluxes have to be treaded very carefully: They have to be wrong, since the gas exchange of CO2 heavily depends on the pressure gradient in CO2 between atmosphere and ocean.

**The choice to hold $CO_2$ constant at 270 ppm will indeed have an effect on DIC(dis). And of course, as George Box famously noted, all models (including this one) are wrong. However, as discussed on page 11 in lines 22-25, the atmospheric $CO_2$ acts primarily on the saturation concentration, while the effect on the disequilibrium fraction is expected to be small. The "minor, non-linear effects" here refer to the fact that air-sea gas exchange is itself a function of DIC(dis), as one term in the calculation of the piston velocity is the departure from equilibrium between the atmosphere and surface ocean with respect to $CO_2$ (using an atmospheric value of 270 ppm), as the reviewer has asserted. We recognize that this is an important point to address in more detail, and we will therefore provide a short quantitative estimate of the magnitude of this effect, along with additional clarification earlier in the paper (page 6, section 4: Discussion).**

**In addition to the fact that the effect is likely to be small, we note that the fact that the carbon cycle is equilibrated with 270 ppm in all cases helps to provide a cleaner comparison between simulations. The carbon cycle has many moving parts, and by holding the atmospheric $CO_2$ constant we provide a simpler picture of mechanistic differences. We would note that the value of the disequilibrium component will also depend on the base state of the soft tissue pump strength, ocean circulation, and alkalinity distribution, none of which are perfectly simulated by any model that we are aware of. Our work is not meant to be a definitive quantification of the disequilibrium component, rather, we hope to illustrate its conceptual importance (which has been overlooked) and show the basic controlling mechanisms.**

4. The authors have chosen to keep atmospheric CO2 fixed, so they calculate changes in oceanic DIC only as a function of prescribed CO2 (always 270 ppm), which ignores dynamic aspects of the gas exchange, that largely depend on the surface ocean-atmospheric difference in pCO2. This is a significant simplication, which reduces the significance of the quantification of the process separation a lot. It implies, that atmospheric CO2 concentration is not a dynamic part of the carbon cycle analysis anymore. CO2 is nevertheless varied, but only to generate different background climatologies, implying only the radiative forcing of CO2 is used here. I therefind find the description of all scenarios and results highly confusing, they should not be defined by the prescribed atm CO2 value, because this is not considered in the carbon cycle change, but by the resulting global annual mean surface temperature changes, ΔT. I therefore expect, that the authors, (i) calculate ΔT, probably with respect to their control simulation (probably the one with CO2 = 270 ppm), and (ii) use ΔT when describing the scenarios, in the text, in Table 1, and in the Figures (e.g. x axis of Fig 1, 4). Since changes in

obliquity, precession, and land ice sheet might also change ΔT, they might also be more specific and call this ΔTCO2, but they might then also analyse the temperature change related to these other processes. If they then plot results in Fig 1 as function of Delta T, a lot of the various scenarios with similar CO2, but different other boundary conditions might then separte in ΔT, and might be easier to be identified. Right now Fig 1 is an mess, with various symbols plotted on top of each other. If thss step does not improve figure 1, the authors might also consider to plot Fig 1 as bar charts, where different scenarios by definition are plotted NEXT and not ON TOP of each other.

**Thank you for the suggestion; we can see that the radiative-$CO_2$ labeling was a major source of confusion for both reviewers, given that it is independent of the $CO_2$ used for air-sea exchange. We will relabel the axes on these plots in terms of the associated change in mean surface air temperature (ΔT). We also appreciate the suggestion to alter the plot style, but substituting a bar plot for figure 1 (as in figure 2) would reduce the amount of information that we are able to convey by eliminating the x-axis and thereby the depiction of the average DIC concentration as a function of ΔT, so we would prefer to retain this plot style, which is more easily legible when plotted vs. temperature.**

5. Anomalies on mean ocean DIC are analysed given in μmol/kg. However, I would find it much more helpful, if the amount of carbon taken up by the ocean would be given in terms of PgC (= GtC), which should be transferable easily (if the mean density of water and the volume of the ocean are known). Maybe, if the authors insist on their view on the system (μmol/kg), they might simply add a 2nd y-label (right-handside) with PgC. This would help a lot, since it is not clear to me, if the setup of the climate model in the LGM mode (more land ice) would also imply less ocean volume, which would directly affect the concentration of DIC in μmol/kg, but not total amount ocean C in PgC. The discussion how much a change in DIC would change atm CO2 (Discussion, page 6) can be simplified a lot by stading the change in oceanic DIC in PgC.

**Thank you for the suggestion. We will add a second y-axis indicating the change in DIC in PgC. The range of the changes in the ocean volume across the different simulations is less than 0.5%, so the changes in DIC concentration and oceanic C inventory are approximately proportional.**

6. In the text various times a change in carbon due to a change in ocean circulation is seen (e.g. page 5, line 26, line 29, page 6, line 2), however, the ocean circulation state is never described in the draft. The reader does not know which of the scenarios has a slower ocean overturning. If this relationship should be kept in the text, some further details (ocean circulation analysis) is needed. My impression is, this might be found in the draft Galbraith + de Lauvergne (submitted), but this is not accessible. So, either the resubmission of this paper has to wait for the other paper, or some of these analysis need to be repeated here.

**Indeed, this is a key aspect of the manuscript submitted by Galbraith and de Lavergne. Because this is a journal without an open discussion (Climate Dynamics), it is, as the reviewer points out, not yet available. However, we anticipate that it should be accepted before we resubmit this manuscript. We will also add a short summary of the overturning circulation in the different scenarios to the results section.**

7. I disagree with the potential usability of the quantification of the soft tissue pump on page 6, implying that the equations might be used as simplification in more complex models.

First. eq 6 should be deleted, since the change in atm CO2 can only be quantified once atm CO2 is calculated dynamically. Second, the ocenic DIC uptake (eq 5) has also as major weakness to deal with the overall setup, that include constant and identical CO2=270 ppm, which implies, that quantification have to be discussed with care. Here, they are taken as given quantification of the soft tissue pump, which might even be used elsewhere. I do not think, this is the case for the reason given above. Furthermore, the sentence, page 7, line 22 "In contrast, the model suggests that greater ocean ventilation rates in the glacial state would have led to reduced global DICsoft." and the following discussion is coming from nowhere and is not supported with any data. We know nothing on the ocean ventilation stage so far. Later-on, in section 4.7 it is argued, that interactive CO2 would only lead to minor effect. I strongly disagree, since the atmosphere-surface ocean gas exchange is a function of the pCO2 difference of both.

**We include equation 6 in order to illustrate the approximately linear dependence of changes in $CO_2$ due to the soft tissue pump and the product of global export and ideal age of the ocean. As discussed above, the major differences in DIC(total) due to setting atmospheric $CO_2$ to 270 ppm should be in the DIC(sat) term. Our calculations are based on a prescribed atmospheric value of CO2 of 270 ppm; thus, using the equation R = ($\Delta CO_2/CO_2$)/($\Delta DIC/DIC$), we simply substitute equation 5 for $\Delta DIC(soft)$, the global average DIC(total) for DIC, $CO_2$ = 270 ppm and R = 10, we derive equation 6. But we do not mean to use this to ignore the intricacies of this or any other general circulation model; our aim is simply to demonstrate how these three variables appear to be related in these scenarios, in order to improve the conceptual understanding. Because it is common the think about carbon reservoir changes in terms of atmospheric $CO_2$, we feel it is valuable to leave equation 6 as it is, but we will add a discussion to this section to clarify that this is indeed a simplification. We would also point out that Ödalen et al. (2017) independently made an almost identical simplification.**

8. Whenever a statement is made, which variable change how much, which is also find in a figure, please include a reference to this figure.

**Thank you for this suggestion; we will add these references.**

9. The meaning of the four separations of DIC (DICsat, DICdis, DICsoft, DICcarb) are explained in the abstract, but not again in the introduction. I believe, this should be repeated around line 10 (page 2), to make the main text independent from the abstract.

**Thank you for this suggestion; we will add this to the introduction.**

10. page 2, line 24: "DICsoft depends ... on the flushing rate of the deep ocean which clears out accumulated DICsoft". This is a bit sloppy and not correct: DICsoft depends on the ocean circulation as a whole, the surface to deep ocean transport also transports it to the deep ocean.

**We agree and will alter this phrasing.**

11. page 3, line 11: "preformed DIC" is not explained/defined her, onlxy later on page 4.

**We will add an explanation of preformed DIC to the introduction along with the description of the DIC decomposition (comment #9).**

12. page 3, line 25f: "Here, we use a fully-coupled general circulation model (GCM) to investigate the potential importance of DICdis in altering air-sea CO2 partitioning on long

timescales." This aim of the paper is not given as such in the abstract, but should be contained there.

**We will add this to the abstract as suggested.**

13. page 3, line 30: "prescribe a constant CO2": It is not only constant in individual experiments, it is also identical in all experiments as 270 ppm. This should be clarified here directly, because it implies, that the atm CO2 is only a driver of climate, but not of carbon cycle changes.

**We will clarify this at this point in the text.**

14. page 4, line 5: "static land and ice sheet". Please explain, or should this read "static land ice sheets"?

**Thank you for catching this; the list should read "…a sea ice module, static land, *and static* ice sheets." This will be changed in the text.**

15. page 4, line 10: The paper Galbraith and de Lavergne (submitted) is missing in the reference list, but need to be included there, at best with a link to an accessible version. If not, maybe introduce this information only, once the paper is accessible somewhere, eg in the next iteration of the paper.

**We will include this information in the bibliography. As noted (cf. #6), we expect that that paper will be accepted for publication before this manuscript is resubmitted and in any case before publication.**

16. page 4, experiments: As already given above, the different CO2 levels should be transfered into ΔT. Also give a brief reasoning for your choices of CO2 here, e.g. why is the reference CO2 270 ppm, and not 278 ppm, as usual, why 607 and 911 ppm?

**The reference $CO_2$ of 270 ppm is used as a simplification for interglacial $CO_2$. The radiative forcing is proportional to $\ln([CO_2])$; thus, 405, 607 and 911 ppm are chosen as linear increments above 180 and 270 ppm (the former approximately represents glacial $CO_2$). This rationale will be added to the text on page 4, section 2.2.**

17. page 4, experiments: Orbital parameters: Eccentricity seemed to have not changed, but nevertheless, state its values, and for which climate state it is typical. Also state, typical values of obliquity and precession for today and LGM.

**This is correct: eccentricity is constant in all of these simulations. This as well as the values for obliquity and precession for modern and LGM conditions will be added to page 4, section 2.2.**

18. page 4, line 17: Iron fertilization: Did I get it right, that the glacial dust fields only change iron availability, but not the radiative forcing? Please clarify. Furthermore, the field in the cited paper (Nickelsen and Oschlies 2015) are taken from Mahowald et al 2006, which should be state here. Also be aware (and potentially discuss), that, at least to my knowledge, more recent LGM dust fields of the Mahowald groud differ to the dust field published in 2006 (especially in the high-latitudes), (e.g. Albani et al 2012 (Clim Dyn), Albani et al 2016 (GRL), which might have an impact on Southern Ocean iron fertilization.

**This is correct: the radiative forcing is not adjusted based on the dust field. This will be clarified in the text. We will also add the Mahowald et al. (2006) reference to page 4, line 17 and note the disagreement with more modern reconstructions.**

    19. page 5, line 18: "standard deviation of only 4 μmol/kg". I do not see any error bars in Fig 1, and in DICcarb the values vary by 20 to 35 μmol/kg. Something is wrong here.

**As stated in the text (page 5, line 18: "over the entire range of $CO_2$ values"), this standard deviation is not an error bar for individual simulations, which are simply run to equilibrium, but rather the standard deviation of the values of DIC(carb) among all simulations. This will be clarified in the text.**

    20. page 5, line 23: "under nutrient depletion experiment". Which one would that be? I only see Fe fertilization expleriments, which would be the opposite.

**This refers to the experiments run by Ito and Follows (2013), as indicated in the text. We did not conduct nutrient depletion experiments but, as noted, rather the opposite, in order to investigate the same dependence. We will clarify the phrasing here.**

    21. page 5ff: I suggest to combine "3 Results" and "4 Discussions" into "3 Results and Discussions" since sec 3 so far is pretty short and sec 4 also includes results.

**We plan to restructure these two sections in the revision; thank you for the input.**

    22. page 6, line 30: Define "ideal age".

**The ideal age tracer is a conservative tracer that is set to 0 in the surface ocean and increases by 1 for each model year in the ocean interior. This definition will be added to the first discussion of this tracer in the main text (page 5, line 21).**

    23. page 6, line 31,32: "quantitative strength of the relationship is striking". It is, what I would expect. The interesing part for me would actually be, to see the individual contribution of the two variables combined in Figure 3, so please also plot "global export" and "age" and define both precisely (are they averaged, if yes, over which results?).

**The correlation with DIC(soft) and each of these parameters individually is lower than the correlation of DIC(soft) and the product of global export and age, which is why only the latter is presented here. Indeed, global export and age are individually averaged spatially over the global ocean for each simulation, and then the product of these two averages is taken. This will be clarified in the text (page 6, lines 29-30). We will also add the two additional panels, as suggested.**

    24. page 7, eq 5,6: These regressions equations are of very limited use, since atm CO2 has been kept constant at 270 ppm, and the gas exchange is a linear function of the atmosphere-surface ocean CO2 difference.

**As in comment #7, these equations are given simply to illustrate the apparent linear relationship between global export * ocean age and DIC(soft), which can roughly be translated into a change in atmospheric $CO_2$ related to the soft tissue pump. Therefore, although coefficients in these equations are only derived empirically, we would argue that they add to the text by showing the relationship depicted in figure 3 mathematically.**

    25. page 7, line 22: We know nothing on ocean ventilations rates yet.

**As in comment #6, we will add a brief summary of the central results of the Galbraith and de Lavergne paper, which will include the results of ocean circulation and ventilation in the different simulations.**

    26. page 8, line 2: Here surface and deep ocean are split at 1 km depth, but in Fig 5 at 500 m depth, Please be consistent.

**Thank you for catching this inconsistency, which reflects a change in the choice of boundary in earlier versions of the manuscript; we will change this to consistently use 1 km for the boundary between the surface and deep ocean.**

    27. page 8, Eq 7: Here, the Southern Ocean contributen is termed fAABW, while in Eq 10 it is termed fSO. Please be consistent.

**We consciously draw a distinction between $f_{SO}$ (the fraction of water originating from the surface south of $30^oS$) and $f_{AABW}$, which is not explicitly traced in the model but would be the true fraction of AABW. Similarly, we use $f_{NAtl}$ and $f_{NADW}$ to refer to the modeled approximation and true fractions of NADW, respectively. We will clarify this in the text (page 8, lines 7-8).**

    28. page 8, line 15: "hot and cold climate state": Be more precise what this means in terms of scenarios. This sentence probably refers to Fig 4, but no reference to it was given.

**"Hot" and "cold" climate states refer to the extreme radiative forcing scenarios ($CO_2$ at 911 and 180 ppm, respectively). We will clarify this in the text and add a reference to figure 4 on page 8, line 15.**

    29. page 8: I suggest to combine the to subsection 4.2 and 4.3 to one subsection.

**We will restructure these sections.**

    30. page 8, line 26: "when deep convection is occuring" When does it occur and where, scenario?

**Deep convection in the model is sporadic in the simulations. The simulations with deep convection are those shown in figure 4 b. In the Southern Ocean, this includes all simulations; in the North Atlantic, this is all simulations at $CO_2 = 180$, 220, and 911 ppm, 1 (of 4) at $CO_2 = 270$ and 405 ppm, and 3 (of 4) at $CO_2 = 607$ ppm. We will allude to this in the text (page 8, line 26).**

    31. page 8, line 27: "... ventilation rates are high at both the cold and hot extremes". Show plot.

**We will include a plot showing this.**

    32. page 9, line 9: O2;dis is not defined (I believe). You might add why O2 equilibrates an order of magnitude faster than CO2 (no bottleneck of the carbonate chemistry during oceanic uptake).

**We will define $O_2$(dis) on page 9, line 6 (equivalent to DIC(dis), it is equal to the departure from equilibrium of $O_2$ in the surface ocean with respect to the atmosphere and advected into the ocean as a conservative tracer). Additionally, we will briefly discuss why it is able to equilibrate approximately an order of magnitude faster than DIC, as stated on page 9, lines 6-7.**

33. page 9, line 11: "O2;dis in the SO is as high as 100 µmol/kg". I believe this is wrong, should be "as low as -100 µmol/kg", and the reference to the figure is needed (Fig 9). O2;dis seems to be largely anticorrelated to DICdis, but this was never mentioned as such.

**We will change this in the text to read "The magnitude of $O_2$(dis) in the SO is as high as 100 µmol/kg" and add the reference to figure 9. We will also mention the anticorrelation.**

34. page 9, line 21: "reduced sensitivity of export" to what?

**We have written "reduced sensitivity *to* export;" this refers to the relationship between DIC(soft) and global export (cf. page 9, line 20). We will clarify this in the text.**

35. page 9, last line: I do not undertstand why the two effects of iron fertilization and remineralization rate should be linearly additive.

**We agree that this is not obvious from first principles, but it appears to be the case in this model. This is evidenced by the fact that DIC(Fe fert)-DIC(no Fe fert) for each of the DIC components is approximately the same using both remineralization rates under pre-industrial conditions, as shown in the first two panels of figure 2. We will clarify this in the text (page 9, lines 30-32).**

36. page 10ff: Change $NO_3$ in $NO_3^-$

**We will make this change.**

37. page 10, line 13: What happens to you framework, if N:C ratios are not constant, as for example in postulated by Geider et al. (1998)?
Geider, R. J.; MacIntyre, H. L. Kana, T. M. A dynamic regulatory model of phytoplanktonic acclimation to light, nutrients, and temperature, Limnology and Oceanography, 1998, 43, 679-694.

**This would indeed the possibility of calculating DIC(soft) from $NO_3^-$; i.e. equation 8 would no longer hold. The RMSE of the DIC(dis)+DIC(soft) is 5.2 µmol/kg; to achieve this error through changes in the N:C ratio only, this ratio would have to vary by more than 6%, which Geider et al. (1998) suggest is possible. However we still feel that this is an improvement on using preformed $PO_4^{3-}$, given that P:C varies much more dramatically (Geider and La Roche, 2002; Galbraith and Martiny, 2015). We will add a short discussion of this point to the text at the end of this section (page 11, line 18).**

38. page 10ff: Unified framework: Here it is argued that the NO3;pre can be split in the contributions from Southern Ocean (SO), North Atlantic (NAtl) and North Pacific (NPac), later-on the argument is made that the North Pacific can be neglected due to missing deep convection. The contributions to DICdis;deep early in the draft (sec 4.2) was only split between NAtl and SO, but not NPac. Please be consistent in both approaches.

**We will state earlier on page 10 that the contribution from the North Pacific is negligible and remove it from equation 10.**

39. page 11, Eq 14 and 15: I have the feeling the factors Vi=Vtotal which are included in Eq 14 are missing in Eq 15, but should still be included here.

**These are absorbed into the individual terms of global $NO_3^-$, $NO_3^-$(den), DIC(dis), and $NO_3^-$(pre). For example, $NO_3^-$(global) = V(deep)/V(total) * $NO_3^-$(deep) + V(upper)/V(total) * $NO_3^-$(upper). Thus, the volume terms drop out of equation 15.**

40. page 11, line 17: RMSE: How has this been found? Is this the mean difference to the 1:1 line in Fig 10?

**Yes, this is calculated from the modeled values of the different components (global average in each simulation) compared to the parametrized values (calculated from equation 15). We will clarify this in the text (page 11, lines 16-18).**

41. page 11, Section 4.7: I think this section might be called "General discussion". I do not think the naming of "DICdis nadir" is helpful here, pleasxe consider other wording.

**We will revise and rename this section.**

42. page 12, lines 7-13: This paragraph is highly speculative and with the given support not justified. We know nothing on AABW formation for different climate state and the fixed, constant atmospheric CO2 boundary condition hinders in my view that such bold hypothesis are made based on the made analysis.

**Indeed, this paragraph is deliberately speculative. This is stated in the text ("We do not claim that this soft upper limit was significant, but simply propose the possibility as a hypothesis that can be tested"). As such, we find it a useful contribution to the paper, as it points to a testable hypothesis for future work.**

43. page 13, line 13: "ratio of remineralized to UTILIZED O2"

**Thank you; we will add "utilized" to the text (page 13, line 13).**

44. page 13, line 18ff: Preformed alkalinity (alkpre) is defined as total alkalinity at the surface. However, alkpre is then calculated from T, S, NO3, PO4. Does this imply, total alkalinity is not followed as an independent tracer in the carbon cycle of the ocean in the model? If not, this need some explanation, since normally the full carbon cycle needs 2 variables to be fully prognostic, typically the conservative tracers DIC and ALK are taken for that (from which pH, CO2, CO3, HCO3 are then calculated. Please clarify, and explain.

**Preformed alkalinity was not included in the simulations. Therefore, as described at this point in the text, performed alkalinity has been a posteriori reconstructed as a linear function of temperature, salinity, and nutrients, similar to the approach used by Bernardello et al. (2014). This will be clarified in the text (page 13, line 18).**

45. page 14: Eq A6 is trivial, its another version of Eq A1.

**We will remove equation A6 from the text.**

46. Figures: Make plots larger to have larger font size, include sub-plot names (eg Fig 1a,bc), check if units are always given (missing in Fig 11). In all figures each caption needs to have the full explanation of what is seen here (and not referring to Fig 1 as done so far).

**We will increase the font sizes, include subplot labels, and write the marker key (as in figure 1) in each of the plots using this scheme (figures 3, 4, 5, and 10).**

47. Fig 1: Plot all 5 subfigures on top of each other, planing, that this will fill 1 column in the final layout text, change x axis to ΔT, consider plotting it as bar charts, clarify what you mean in the caption with changed seaonality (probably a change in pressession).

**We will plot the five subplots vertically and change the x-axis to ΔT. As above (cf. #4), we prefer to retain this plot style to demonstrate the relationship between DIC and radiative**

**forcing in this model. Indeed, "seasonality" refers to precession, which we will clarify in the caption.**

48. Fig 2: If results are averages from 4 runs (as said here) give mean and error from averaging.

**Indeed, this assertion is correct. As the means are given (these are the values plotted), we will include the standard deviations of the respective four runs as error bars and clarify this in the caption.**

49. Fig 3: X axis: "Global export times age". "Global export" here implies only export of organic C or also CaCO3 export?

**This refers to export of organic C only; we will clarify this in the figure caption.**

50. Fig 4: Why do you take DICdis at 100m water, and not the average over the mixed layer?

**We show DIC(dis) at 100 m depth given that we are using annual mean concentrations, and there is a strong seasonal cycle of mixed layer depths at high latitudes. The depth of 100 m was chosen keeping in mind that polar deep-water formation regions tend to have strong haloclines, and 100m tends to be below the summer mixed layer but within the winter mixed layer, and is therefore relatively representative of the mixed layer concentrations when deep waters are forming.**

51. Fig 6,8,9,11: Define scenarios "glacial" and "interglacial". Are these averages over several scenarios? Unit is missing in Fig 6.

**These are single scenarios where the $CO_2$, obliquity, precession, and ice sheet configuration are 180 ppm, 22º, 90º, and LGM for the "glacial" scenario and 270 ppm, 24º, 90º, and pre-industrial for the "interglacial" scenario, respectively. We will clarify this in methods and in the figure captions and will add the units of DIC(dis) (µmol/kg) to the caption of figure 6.**

52. Fig 7: I find this figure, highly confusing (not clear what lines represent), and not necessary at all.

**We agree and will remove this figure.**

53. Fig 10: Caption: Please reduce text and refer to where the Eq is found in text, e.g. lower plot y axis follows Eq 15. I am not sure for the other 2 plots there is a complete Eq contained in the theoretical framework, if not, please extend.

**We will cite equations 11, 12, and 15 in the caption and reduce the caption length by referring to the appropriate section of the text for details (section 4.6). As we note in the text, we only expand the equations for the deep ocean (page 11, lines 1-2), but the full equations for DIC(dis) and DIC(soft) follow analogously to equation 13.**

54. Fig 11: I am not sure, this Fig is necessary.

**We find this important to show where the parametrization used for preformed alkalinity induces errors in the DIC(dis) analysis (cf. #44), as this is the main source of uncertainty. Thus, we would prefer to retain this figure in the appendix.**

   As noted by Anonymous Reviewer 1, this study uses a fixed and constant atmospheric CO2 mixing ratio of 270 ppm in the numerical model experiments. While I see nothing intrinsically wrong with this approach, the manuscript as it is does not adequately describe the impacts of this choice on their results.
   For example, consider Figure 1 and accompanying text: On p5 lines 15-17, the manuscript states how a constant $CO_2$ for gas exchange of 270 ppm is used. Then, the manuscript states how the largest changes in DIC_sat is driven by the changes in $\ln(CO_2)$. Figure 1 does indeed plot DIC_sat against CO_2 with CO_2 shown to vary.
   I do not understand this figure or the text: precisely how is DIC_sat a function of $\ln(CO_2)$ if a constant CO_2 of 270 ppm is used for gas exchange. If a constant CO_2 of 270 ppm is used for gas exchange in the numerical simulations, then surely the CO_2

cannot be changing on the axes in figure 1. Also, DIC_sat should always be calculated relative to a CO_2 of 270 ppm, and so DIC_sat will not change with ln(CO2).

The issue recurs in Figure 2, the caption to which indicates that some experiments are run at 180 ppm CO_2 concentration.

This confusion is critical for assessment of the manuscript in its current form (e.g. also see major point 2, which derives from this).

Note that there are other studies in this topic that have dealt with similar issues well. For example, Marinov et al (2008) (cited by this manuscript) uses GCM simulations with a fixed temperature for air-sea gas exchange, but a varying dynamical temperature for ocean circulation. In that manuscript, the issue is well described and the findings are clear.

Is the issue here that one CO_2 is used for the radiative forcing of climate, but another CO_2 is used for air-sea gas exchange?

I cannot tell precisely with the manuscript in its present form. If this is the case, then all mentions of CO_2 in ppm [and ln(CO2)] that are different from 270ppm could be converted to radiative forcing (in W/m^2) with respect to CO_2 = 270 ppm. For example, CO_2 = 180ppm would be re-defined as 'Glacial radiative forcing' (or a numerical radiative forcing of _ -2.2 W/m^2).

For example: P11 Line 20 to line 24 reads: "The model simulations show a clear minimum DIC_dis at intermediate CO_2 (270-405 ppm). : : : the CO_2 driving gas exchange : : : is held fixed at 270 ppm."

This could be changed to something like: "The model simulations show a clear minimum DIC_dis at intermediate Radiative Forcing (0 – 2.2 Wm-2). : : : the CO_2 driving gas exchange : : : is held fixed at 270 ppm."

I am currently confused by the way this issue is written about in the study. Please clarify.

**Indeed, different $CO_2$ levels are used for the radiative forcing, which is varied from 180 to 911 ppm in these simulations, and $CO_2$ used for the biogeochemistry, including air-sea gas exchange, which is always 270 ppm. Given the similar confusion raised by Reviewer 1, we will refer only to the different climate forcings in terms of the mean surface temperature ($\Delta T$). We feel this will alleviate a great deal of misunderstanding.**

2. DIC_sat and DIC_dis definitions:
   There are two ways of defining DIC_sat and DIC_dis in the present-day system of rising CO_2, or over different periods when CO_2 has changed in the past.

   Firstly, you can define both DIC_sat and DIC_dis relative to some fixed atmospheric CO_2 concentration (such as the preindustrial). Or secondly, you can define them relative to the current atmospheric CO_2 concentration at a particular point in time (for example it would be relative to 180 ppm at the LGM, or _400ppm in the present day).

   This choice makes a big difference. Consider the present day: if DIC_sat is defined relative to present day atmospheric CO_2 then DIC_dis is small in the surface ocean and negative in the deep ocean. However, if DIC_sat is defined relative to preindustrial CO_2 then DIC_dis is positive in the surface ocean and _ zero at depth.

   In the original discussions of DIC_sat (p2, lines 14-17) and DIC_dis (p3, lines 14-34), it is unclear whether a fixed or rising CO_2 concentration will be used to define DIC_dis and DIC_sat.

   It eventually becomes clear (I think) that in this study, the DIC_sat is calculated relative to a fixed CO_2 of 270 ppm (page 4, line 15). However, the point is only made when discussing the numerical model set up.

A clearer indication of how DIC_sat and DIC_dis are treated in this study from the outset is required. Especially given the confusing 'fixed but changing' CO_2 issue from my other major concern. If the experiments are run with CO_2=180ppm, and DIC_sat is defined relative to 270ppm then this will have a large impact on the results.

**We agree this is potentially confusing. We will add text to the introduction to precisely define the terms, as we use them. We will also add a new figure to illustrate these important concepts clearly. In our usage, both DIC(sat) and DIC(dis) are determined only in the surface layer and are propagated into the interior by mixing and advection. Thus, under a transient change of $CO_2$, the surface values would evolve as the $CO_2$ changes, and the values propagated into the interior would follow.**

**The devil's in the disequilibrium:  multi-component analysis of  dissolved carbon  and  oxygen changes under a broad range of forcings in a general circulation model**

Sarah Eggleston[1,*] and Eric D. Galbraith[1,2,3]

[1]Institut de Ciència i Tecnologia Ambientals (ICTA), Universitat Autònoma de Barcelona, 08193 Barcelona, Spain
[2]Institució Catalana de Recerca i Estudis Avançats (ICREA), Pg. Lluís Companys 23, 08010 Barcelona, Spain
[3]Department of Earth and Planetary Science, McGill University, Montreal, Quebec H3A 2A7, Canada
[*]Now at: Laboratory for Air Pollution & Environmental Technology, Empa, Überlandstrasse 129, 8600 Dübendorf, Switzerland

*Correspondence to:* S. Eggleston (sarah.eggleston@gmail.com)

**Abstract.**  The complexity of dissolved gas cycling in the ocean presents a challenge for mechanistic understanding and can hinder model intercomparison. One helpful approach is the conceptualization of dissolved gases as the sum of multiple, strictly-defined components. Here we decompose dissolved inorganic carbon (DIC) into four components: saturation ($DIC_{sat}$), disequilibrium ($DIC_{dis}$), carbonate ($DIC_{carb}$) and soft tissue ($DIC_{soft}$).  The cycling of dissolved oxygen is simpler, but can still be aided by considering $O_2$, $O_{2_{sat}}$ and $O_{2_{dis}}$. We explore changes in  these components within a large suite of simulations with a complex coupled climate-biogeochemical model, driven by changes in astronomical parameters, ice sheets and  radiative forcing, in order to explore the potential importance of the different components to ocean carbon storage on long timescales. We find that both $DIC_{soft}$ and $DIC_{dis}$ vary over a range of 40 $\mu$mol/kg in response to the climate forcing, equivalent to changes in atmospheric $p$CO$_2$ on the order of 50 ppm for each.  The most extreme values occur at the coldest and intermediate climate states. We also find significant changes in $O_2$ disequilibrium, with large increases under cold climate states. We find that, despite the broad range of climate states represented, changes in global $DIC_{soft}$ can be  quantitatively approximated by the product of deep ocean ideal age and the global export production flux. In contrast, global $DIC_{dis}$ is dominantly controlled by the fraction of the ocean filled by Antarctic Bottom Water (AABW). Because the AABW fraction and ideal age are inversely correlated  among the simulations, $DIC_{dis}$ and $DIC_{soft}$ are also inversely correlated, dampening the overall changes in DIC. This inverse correlation could be decoupled if changes in deep ocean mixing were to alter ideal age independently of AABW fraction, or if independent ecosystem changes were to alter export and remineralization, thereby modifying $DIC_{soft}$. As an example of the latter, we show that iron fertilization causes both $DIC_{soft}$  and $DIC_{dis}$ to  increase and that the relationship between these two components depends on the climate state. We propose a simple framework to consider the

global contribution of $DIC_{soft} + DIC_{dis}$ to ocean carbon storage as a function of the surface preformed nitrate and $DIC_{dis}$ of dense water formation regions, the global volume fractions ventilated by these regions, and the global nitrate inventory. ~~More extensive sea ice increases $DIC_{dis}$, and when sea ice becomes very extensive it also causes significant $O_2$ disequilibrium, which may have contributed to reconstructions of low $O_2$ in the Southern Ocean during the glacial. Global $DIC_{dis}$ reaches a minimum near modern $CO_2$ because the AABW fraction reaches a minimum, which may have contributed to preventing further $CO_2$ rise during interglacial periods.~~

*Copyright statement.* Both authors accept the licence and copyright agreement.

**1 Introduction**

The controls on ocean carbon storage are not yet fully understood. Although potentially very important, given the large inventory of dissolved inorganic carbon (DIC) the ocean contains (currently 38,000 Pg C vs. 700 Pg C in the pre-industrial atmosphere), the nuances of carbon chemistry, the dependence of air-sea exchange on wind stress and sea ice cover, the intricacies of ocean circulation, and the activity of the marine ecosystem all contribute to making it a very complex problem. The scale of the challenge is such that, despite decades of work, the scientific community has not yet been able to satisfactorily quantify the role of the ocean in the natural variations of $p$CO$_2$ between 180 and 280 ppm that occurred over ice age cycles. This failure reflects persistent uncertainty that also impacts our ability to accurately forecast future ocean carbon uptake  (Le Quéré et al., 2007; Friedlingstein et al., 2014).

In order to help with process understanding,  Volk and Hoffert (1985) proposed conceptualizing ocean carbon storage as consisting of a baseline surface-ocean average, enhanced by two "pumps" that transfer carbon to depth: the solubility pump, produced by the vertical temperature gradient, and the soft-tissue pump, produced by the sinking and downward transport of organic matter. This conceptualization proved very useful, but it fails to deal explicitly with the role of spatially- and temporally-variable air-sea  exchange, and cannot effectively address changes in ocean circulation. A number of other conceptual systems have been employed (e.g. Broecker et al., 1985; Gruber et al., 1996), both for considering natural changes in the carbon cycle of the past and the anthropogenic transient input of carbon into the ocean.

Here, we use the decomposition laid out by Williams and Follows (2011), with the small change that we consider only DIC, rather than total carbon. This theoretical framework defines four components that, together, add up to the total DIC: saturation ($DIC_{sat}$), disequilibrium ($DIC_{dis}$), carbonate ($DIC_{carb}$) and soft tissue ($DIC_{soft}$) (Ito and Follows, 2013; Bernardello et al., 2014; Ödalen et al., 2018). The first two components are "preformed" quantities ($DIC_{pre} = DIC_{sat} + DIC_{dis}$), i.e. they are defined in the surface layer of the ocean and are carried passively by ocean circulation  into the interior. In contrast, the latter two are equal to zero in the surface layer and accumulate in the interior due to biogeochemical activity  (see fig. 1). Note that the four components are diagnostic quantities

only, intended to aid in understanding mechanisms and clarifying hypotheses, and do not influence the behavior of the model (although they can be calculated more conveniently by including additional ocean model tracers, as described in Methods).

Saturation DIC  is simply determined by the atmospheric $CO_2$ concentration and its solubility in seawater, which is a function of ocean temperature, salinity, and alkalinity. For example, cooling the ocean will increase $CO_2$ solubility, thereby leading to an increase in $DIC_{sat}$. Given known changes in temperature, salinity, alkalinity, and atmospheric $pCO_2$, the effective storage of $DIC_{sat}$ can be calculated precisely.

 At the ocean surface, primary producers take up DIC. The organic carbon that is formed then sinks or is subducted (as dissolved or suspended organic matter) and is transformed into remineralized DIC within the water column (a small fraction is buried at depth). Here we define $DIC_{soft}$ as that  which accumulates through the net respiration of organic matter below the top layer of the ocean (in our model, the uppermost 10 m). Thus, $DIC_{soft}$ depends both on the export flux of organic matter, affected by surface ocean conditions including  nutrient supply (Moore et al., 2013; Martin, 1990), and the ocean circulation as a whole, including the surface-to-deep export and the flushing rate of the deep ocean, which clears out accumulated $DIC_{soft}$ (Toggweiler et al., 2003). The Southern Ocean (SO) is thought to be an important region for such changes on glacial/interglacial timescales, as the ecosystem there is currently iron-limited, and it also plays a major role in deep ocean ventilation  (Mar Assuming a constant global oceanic phosphate inventory and constant C:P ratio, $DIC_{soft}$ would be stoichiometrically related to the preformed $PO_4$  ($PO_{4pre}^{3-}$) inventory of the ocean, where $PO_{4pre}^{3-}$ is the concentration of $PO_4^{3-}$ in newly-subducted waters, and a  passively-transported tracer in the interior. The potential to use $PO_{4pre}^{3-}$ as a metric of $DIC_{soft}$ prompted  very fruitful efforts to understand how it could change over time (Ito and Follows, 2005; Marinov et al., 2008a; Goodwin et al., 2008), though it has been pointed out that the large variation in C:P of organic matter  could weaken the relationship between $DIC_{soft}$ and $PO_{4pre}$ ~~(Galbraith and Martiny, 2015). Dissolved $O_2$ can potentially serve as a better metric of $DIC_{soft}$, given the relatively small variations in the $O_2$:C of respiration compared to the relatively high variability in C:P (Martiny et al., 2013). But Apparent Oxygen Utilization (AOU), typically taken as a measure of accumulated respiration, can be misleading if the preformed $O_2$ concentration differed significantly from saturation (Ito et al., 2004; Duteil et al., 2013). Thus, despite being conceptually simple, $DIC_{soft}$ can be difficult to quantify observationally.~~$_{4pre}^{3-}$ (
[revised manuscript text omitted]
$_{soft}$ ~~= 7.3 $\mu$mol/kg). This is qualitatively the same in the case that the remineralization rate of organic carbon is reduced by 25% (left panel). However, in the high-ventilation runs under glacial-like conditions (right panel), the change in total DIC is somewhat reduced (11.1 $\mu$mol/kg) and $\Delta$DIC$_{soft}$ is small (SO average 4.3 $\mu$mol/kg; global average 2.9 $\mu$mol/kg). DIC$_{dis}$ is also dependent on the ocean circulation, but in the opposite direction: the change is more significant (9.4 $\mu$mol/kg global average) in the glacial-like simulations compared to the pre-industrial simulation (6.4 $\mu$mol/kg). Each of these values rises when considering only the SO (13.4 and 9.0 $\mu$mol/kg, respectively). In both the pre-industrial and glacial-like simulations, changes in DIC$_{dis}$ constitute an important part of the total change. DIC$_{carb}$ is reduced to a small extent~~

by iron fertilization, due to the reduced nutrient content of AAIW/SAMW and consequent decrease in low latitude carbonate production, which raises low latitude surface ocean alkalinity, causing an increase in DIC~sat.~ are negatively correlated.

**4 ~Discussion~**

Simulated changes in $DIC_{dis}$ are of the same magnitude as the $DIC_{soft}$ changes, to which much greater attention has been paid. For a global average buffer factor between 8 and 14 (Zeebe and Wolf-Gladrow, 2001), a rough, back-of-the-envelope calculation shows that a 1 $\mu$mol/kg change in DIC corresponds to a 0.9 – ~1.5~1.6 ppm change in atmospheric $ppCO_2$ based on a DIC concentration of 2300 $\mu$mol/kg and $pCO_2$ of 270 ppm. Thus, the increase in the global average $DIC_{dis}$ in these simulations ~of 16 $\mu$mol/kg (pre-industrial control) to 62 $\mu$mol/kg (LGM-like conditions with Fe fertilization) could have accounted for~ could have contributed more than a 40 ~– 70 ppm difference in atmospheric~ ppm change in the atmospheric $pCO_2$ stored in the ocean during the glacial compared to today. ~While the~ It is important to recognize that the drawdown of $CO_2$ by disequilibrium storage would have resulted in a decrease of $DIC_{sat}$, given the dependence of the saturation concentration on $pCO_2$, so this estimate should not be interpreted as a straightforward atmospheric $pCO_2$ change. Nonetheless, while this is only a first-order approximation and the model biases are potentially large, it seems very likely that the ~desquilibrium~ disequilibrium carbon storage was a significant portion of the net 90 ppm difference.

~Below, we discuss the changes in $DIC_{soft}$ and $DIC_{dis}$ that result from the $CO_2$, orbital and ice-sheet driven climate changes. We then discuss related changes in disequilibrium $O_2$, implications for preformed nutrient theory, and propose a new mechanism that may have helped to prevent the Earth from warming its way out of the ice age cycle.~

**3.1 Climate-driven changes in DIC~soft~**

The biogeochemical model used here is relatively complex, with limitation by three nutrients (N, P and Fe), denitrification and $N_2$ fixation, in addition to the temperature- and light-dependence typical of biogeochemical models. The climate model is also complex, including a full atmospheric model, a highly-resolved dynamic ocean mixed layer, and many nonlinear subgridscale parameterizations, and uses short (< 3 h) timesteps. The simulations we show span a wide range of behaviours, including major changes in ocean ventilation pathways and patterns of organic matter export.

Thus, it is perhaps surprising that the net global result of the biological pump, as quantified by $DIC_{soft}$, has highly predictable behavior. As shown in fig. 4, the global $DIC_{soft}$ varies closely with the product of the global average sinking flux of organic matter at 100 m and the average ideal age of the global ocean. Qualitatively this is not a surprise, given that greater export pumps more organic matter to depth, and a large age provides more time for respired carbon to accumulate within the ocean. But the quantitative strength of the relationship is striking. As demonstrated in fig. 4, global $DIC_{soft}$ is not as well correlated with either of these parameters separately as it is with their produce, "age × export."

It is difficult to assess the likelihood that the real ocean follows this relationship to a similar degree. One reason it might differ is if remineralization rates vary spatially, or with climate state. In the model here, as in most biogeochemical models, organic matter is respired according to a globally-uniform power law relationship vs. depth (Martin et al., 1987). Kwon et al.

(2009) showed that ocean carbon storage is sensitive to changes in these remineralization rates, and this would provide an additional degree of freedom. It is not currently known how much remineralization rates can vary naturally; they may vary as a function of temperature (Matsumoto et al., 2007) or ecosystem structure. As a result, the relationship between $DIC_{soft}$ and  age $\times$ export may be stronger in the model than in the real ocean.

5    Nonetheless, the results suggest that, as a useful first-order approximation, the global change in $DIC_{soft}$ between two states can be given by  a simple linear regression:

$$\Delta DIC_{soft}[\mu mol\ kg^{-1}] = \underline{0.036} m_1 \cdot \Delta(\underline{export[Pg\ C\ y^{-1}] \cdot} age[y] \times export[Pg\ C\ y^{-1}]) \tag{1}$$

or in underline{terms of $p$CO_2}:

$$\Delta CO_2[ppm] \approx m \cdot \Delta(export[Pg\ C\ y^{-1}] \cdot age[y])$$

10

$$\Delta pCO_{2,soft}[ppm] \approx m_2 \cdot \Delta(age[y] \times export[Pg\ C\ y^{-1}]) \tag{2}$$

Note that $m_2$ is a function of the buffer factor and the climate state (atmospheric $pCO_2$ and DIC). Based on the results here, $m_1 = 0.036$ and $m_2 = 0.065, 0.042, 0.029$ for modern (405 ppm $pCO_2$), pre-industrial  (270 ppm) and glacial (180 ppm) conditions, respectively. Note that we have not varied $pCO_2$ in these simulations, so these

15   equations are only meant to illustrate the mathematical relationship observed in fig. 4. This simple meta-model may provide a useful substitute for full ocean-ecosystem calculations, and  should be further tested against other ocean-ecosystem coupled models  with interactive $CO_2$. Note that, as for the disequilibrium estimate above, the soft tissue pump $CO_2$ drawdown would be partially compensated by a decrease in saturation carbon storage, so this will be larger than the net atmospheric effect. In addition, we have not accounted for consequent changes in the surface ocean carbonate chemistry (including changes in the

20   buffer factor).

It is important to point out that the simulated change in $DIC_{soft}$ between interglacial and glacial states  appears to be in conflict with reconstructions of the LGM. Proxy records appear to show that LGM dissolved oxygen concentrations were lower throughout the global ocean, with the exception of the North Pacific, implying greater $DIC_{soft}$ concentrations during the glacial then during the Holocene (Galbraith and Jaccard, 2015). In contrast, the model suggests

25   that greater ocean ventilation rates in the glacial state (fig. 2(a)) would have led to reduced global $DIC_{soft}$. As discussed by Galbraith and de Lavergne (2018), radiocarbon observations imply that the model ideal age is approximately 200 y too young under glacial conditions  compared to the LGM, suggesting a circulation bias that may reflect incorrect diapycnal mixing or non-steady-state conditions. Whatever the cause, if we take this 200 y bias into account, the regression implies an additional 33 $\mu mol\ kg^{-1}$ $DIC_{soft}$ were stored in the glacial ocean. This would bring the simulated

30   glacial $DIC_{soft}$ close to, but still less than, the simulated pre-industrial value.

We propose that the apparent remaining shortfall in simulated glacial $DIC_{soft}$ could reflect one or more of the following non-exclusive possibilities: 1. the model does not capture changes in remineralization rates caused by ecosystem changes; 2. the

model underestimates the glacial increase in the nitrate inventory and/or growth rates, perhaps due to changes in the iron cycle; 3. the ocean was not in steady state during the LGM, and therefore not directly comparable to the  GL simulation; 4. the inference of $DIC_{soft}$ from proxy oxygen records is incorrect due to significant changes in preformed oxygen disequilibrium (see below). If either of the first two possibilities is important, it would imply an inaccuracy in the meta-model derived here.

**3.2 Climate-driven changes in $DIC_{dis}$**

The ocean basins below 1 km depth are largely filled by surface waters subducted to depth in regions of deepwater formation (Gebbie and Huybers, 2011). In our simulations, water originating in the surface North Atlantic, termed NADW, and the Southern Ocean, termed AABW, make up 80-96% of this total deep ocean volume. Thus, to first order, the deep average $DIC_{dis}$ concentration can be approximated by a simple mass balance:

$$DIC_{dis_{deep}} \approx f_{AABW} \cdot DIC_{dis_{AABW}} + f_{NADW}(1 - f_{AABW}) \cdot DIC_{dis_{NADW}} \tag{3}$$

Here, $f_{AABW}$  represents the fraction of deepwater originating in the SO, and $DIC_{dis_{AABW}}$, $DIC_{dis_{NADW}}$ represent the $DIC_{dis}$  concentrations at the sites of deepwater formation (see fig. 5). North Atlantic deep waters form with negative $DIC_{dis}$, reflecting surface undersaturation, while the Southern Ocean is supersaturated ($DIC_{dis} > 0$). These opposing tendencies between NADW and AABW cause a partial cancellation of $DIC_{dis}$ when globally averaged, which makes the disequilibrium component small in the modern ocean. Theoretically, the simulated $DIC_{dis}$ could change either due to changes in $f_{AABW}$ or the end-member compositions. Although the exact values of $DIC_{dis}$ in the two polar oceans vary  among the simulations in response to climate (the reasons for which are discussed in more detail below), these changes are small relative to the consistent large contrast between  $f_{AABW}$ and $f_{NADW}$, so that deep $DIC_{dis}$ is strongly controlled by the global balance of AABW vs. NADW in each simulation (see fig. 6). Global $DIC_{dis}$ becomes much larger when $f_{AABW}$ is larger, similar to the dynamic evoked by Skinner (2009). This is also illustrated by the depth transects of $DIC_{dis}$ in fig. A.

 We estimated the concentration of $DIC_{dis}$ in the regions of AABW and NADW formation, shown in fig. 5(b). The end members vary less significantly than $f_{AABW}$ over the range of simulations, in part due to competing effects of different processes. As discussed in section 3.1, simulations at both the low and high radiative forcing values used show increased AABW production, with a minimum at intermediate values  (fig. 5). The fact that the highest $DIC_{dis_{AABW}}$ occurs at low SAT can be attributed to the rapid formation rate of AABW, while the intermediate-SAT minimum in AABW  volume explains the minimum in global ocean $DIC_{dis}$ (fig. 3).  We note that expanded terrestrial ice sheets shift the ratio of AABW to NADW to higher values, due to their impact on NADW temperature and downstream expansion of Southern Ocean sea ice (Galbraith and de Lavergne, 2018), further increasing $DIC_{dis}$ in glacial-like conditions.

**3.3**

~~Due to the general dominance of AABW in the deep ocean, the concentration of $DIC_{dis}$ in the regions of AABW formation, $DIC_{dis_{AABW}}$, is another important factor determining global $DIC_{dis}$. This varies less significantly than $f_{AABW}$ over the range of simulations, in part due to competing effects of different processes. Surface ocean $DIC_{dis}$ in the Southern Ocean grows in response to upwelling of deepwater, which brings DIC-charged waters to the surface, thus contributing to the carbon~~

[revised manuscript text omitted]

$$NO_{3rem}^{-} = NO_{3global}^{-} - NO_{3pre}^{-} + NO_{3den}^{-} \tag{5}$$

$$NO_{3pre,upper}^{-} \approx f_{SO,upper} \cdot NO_{3pre_{SO,upper}}^{-} + f_{NAtl,upper} \cdot NO_{3pre_{NAtl,upper}}^{-} + f_{NPac,upper} \cdot NO_{3pre_{NPac,upper}}^{-} \tag{6}$$

$$NO_{3pre_{deep}}^{-} \approx f_{AABW} \cdot NO_{3pre_{AABW}}^{-} + f_{NADW} \cdot NO_{3pre_{NADW}}^{-} \tag{7}$$

 Because there is production of intermediate water but no deep convection in the North Pacific, we calculate this mass balance for the upper ocean (above 1 km) and deep ocean separately, dropping the

$$DIC_{soft_{deep}} \approx r_{C:N} \cdot [NO_{3deep}^{-} + NO_{3den,deep}^{-} - (f_{SO,deep} \cdot NO_{3pre_{SO,deep}}^{-} + f_{NAtl,deep} \cdot NO_{3pre_{NAtl,deep}}^{-})]$$

 Pacific Ocean term in eq. 7 for the deep ocean. For brevity, we continue with the derivation for the deep ocean only; the upper ocean follows analogously.

$$\text{DIC}_{\text{soft}_{\text{deep}}} \approx r_{\text{C:N}} \cdot [\text{NO}_{3_{\text{deep}}}^- + \text{NO}_{3_{\text{den,deep}}}^- - (f_{\text{AABW}} \cdot \text{NO}_{3_{\text{pre}_{\text{AABW}}}}^- + f_{\text{NADW}} \cdot \text{NO}_{3_{\text{pre}_{\text{NADW}}}}^-)] \tag{8}$$

Combining with equation 3,

$$\text{DIC}_{\text{dis}_{\text{deep}}} + \text{DIC}_{\text{soft}_{\text{deep}}} \approx r_{\text{C:N}} \cdot (\text{NO}_{3_{\text{deep}}}^- + \text{NO}_{3_{\text{den,deep}}}^-) + f_{\text{SO,deep}} \cdot (\text{DIC}_{\text{dis}_{\text{SO,deep}}} - r_{\text{C:N}} \cdot \text{NO}_{3_{\text{pre}_{\text{SO,deep}}}}^-)$$
$$+ f_{\text{NAtl,deep}} \cdot (\text{DIC}_{\text{dis}_{\text{NAtl,deep}}} - r_{\text{C:N}} \cdot \text{NO}_{3_{\text{pre}_{\text{NAtl,deep}}}}^-) \tag{9}$$

Finally, the global average is computed by summing the volume-weighted values in the upper and deep ocean:

$$\text{DIC}_{\text{dis}_{\text{global}}} + \text{DIC}_{\text{soft}_{\text{global}}} \approx [V_{\text{upper}} \cdot (\text{DIC}_{\text{dis}_{\text{upper}}} + \text{DIC}_{\text{soft}_{\text{upper}}}) + V_{\text{deep}} \cdot (\text{DIC}_{\text{dis}_{\text{deep}}} + \text{DIC}_{\text{soft}_{\text{deep}}})]/V_{\text{total}} \tag{10}$$

Fully expanded, this yields:

$$\text{DIC}_{\text{dis}_{\text{global}}} + \text{DIC}_{\text{soft}_{\text{global}}} \approx \frac{V_{\text{upper}}}{V_{\text{total}}}[r_{\text{C:N}} \cdot (\text{NO}_{3_{\text{upper}}}^- + \text{NO}_{3_{\text{den,upper}}}^-) + f_{\text{SO,upper}} \cdot (\text{DIC}_{\text{dis}_{\text{SO,upper}}} - r_{\text{C:N}} \cdot \text{NO}_{3_{\text{pre}_{\text{SO,upper}}}}^-)$$
$$+ f_{\text{NAtl,upper}} \cdot (\text{DIC}_{\text{dis}_{\text{NAtl,upper}}} - r_{\text{C:N}} \cdot \text{NO}_{3_{\text{pre}_{\text{NAtl,upper}}}}^-)]$$
$$+ \frac{V_{\text{deep}}}{V_{\text{total}}}[r_{\text{C:N}} \cdot (\text{NO}_{3_{\text{deep}}}^- + \text{NO}_{3_{\text{den,deep}}}^-) + f_{\text{SO,deep}} \cdot (\text{DIC}_{\text{dis}_{\text{SO,deep}}} - r_{\text{C:N}} \cdot \text{NO}_{3_{\text{pre}_{\text{SO,deep}}}}^-)$$
$$+ f_{\text{NAtl,deep}} \cdot (\text{DIC}_{\text{dis}_{\text{NAtl,deep}}} - r_{\text{C:N}} \cdot \text{NO}_{3_{\text{pre}_{\text{NAtl,deep}}}}^-)] \tag{11}$$

which can be generalized for any number $n$ of ventilation regions $i$ as:

$$\text{DIC}_{\text{dis}_{\text{global}}} + \text{DIC}_{\text{soft}_{\text{global}}} \approx r_{\text{C:N}} \cdot (\text{NO}_{3_{\text{global}}}^- + \text{NO}_{3_{\text{den,global}}}^-) + \sum_{i=1}^{n} f_i \cdot (\text{DIC}_{\text{dis}_i} - r_{\text{C:N}} \cdot \text{NO}_{3_{\text{pre}_i}}^-) \tag{12}$$

Thus, total carbon storage as soft and disequilibrium carbon (i.e. everything other than $\text{DIC}_{\text{sat}}$ and $\text{DIC}_{\text{carb}}$) varies with the global nitrate inventory, corrected for  accumulated $\text{NO}_3^-$ loss to denitrification, and the difference between $\text{DIC}_{\text{dis}}$ and  $r_{\text{C:N}} \cdot \text{NO}_{3_{\text{pre}}}^-$ in the polar oceans, modulated by their respective volume fractions.

**3.6**

~~The model simulations show a clear minimum of $\text{DIC}_{\text{dis}}$ at intermediate $CO_2$ (270 − 405 ppm). This occurs as a result of the minimum contribution of AABW to the global ocean, and to a lesser extent, a minimum in the $\text{DIC}_{\text{dis}}$ of AABW. A minimum global extent of AABW during interglacials has been documented from $\epsilon_{\text{Nd}}$ (Piotrowski et al., 2005), and it has been shown that~~

~~AABW likely ceased to form entirely for a brief portion of the last interglacial, marine isotope stage 5e (Hayes et al., 2014). Although the model has biases that prevent a strict interpretation of the $CO_2$ and orbital combination at which this nadir is likely to occur, its robust emergence from the relative density control on deep water ventilation volumes makes its existence appear reasonable (Galbraith and de Lavergne, submitted).~~

5   ~~that the lower limit of $CO_2$ over glacial cycles is quite firm and that low $CO_2$ is very common. In addition, they showed that, although the peak values of $CO_2$ during interglacials vary significantly, interglacial $CO_2$ levels are relatively common, suggesting that they are also a preferred state. We propose that the $DIC_{dis}$ nadir would have contributed to preventing a significant $CO_2$ rise above interglacial $CO_2$, since a rise of temperatures above that which gives the $DIC_{dis}$ nadir will increase~~

10 ~~AABW formation and thereby draw down $CO_2$ as $DIC_{dis}$. Of course this would be counteracted by an increase of $DIC_{soft}$, if it were to behave opposite of $DIC_{dis}$, as occurs in our simulations – whether or not this is true depends on unresolved climate dependencies in the marine ecosystem, which currently remains an open question. We do not claim that this soft upper limit was significant, but simply propose the possibility as a hypothesis that can be tested~~Although this nitrogen-based framework avoids the problem of C:P variability, it is not clear how large the effects of variable C:N might be in the real world. This could

15 be a worthy topic for future exploration.

**4   Conclusions**

[revised manuscript text omitted]

$$DIC_{sat} = f(T, S, alk_{pre}, pCO_2) \tag{A2}$$

In this model, $DIC_{soft}$ is proportional to the utilized $O_2$, which is defined as the difference between preformed and total $O_2$, where the ratio of remineralized C to utilized $O_2$ ($r_{C:O_2}$) is 106:150.

30 $$DIC_{soft} = r_{C:O_2} \cdot (O_{2_{pre}} - O_2) \tag{A3}$$

DIC derived from $CaCO_3$ dissolution is proportional to the change in alkalinity, correcting for the additional change in alkalinity due to hydrogen ion addition during organic matter remineralization.

$$DIC_{carb} = 0.5 \cdot [(alk - alk_{pre}) + r_{N:O_2} \cdot (O_{2_{pre}} - O_2)] \tag{A4}$$

Preformed alkalinity, defined as the total alkalinity at the surface and treated as a conservative tracer,  is calculated within the model framework but was not written out during the model runs. Therefore, we have reconstructed this parameter a posteriori for each model year through multilinear regressions as a function of century-averaged salinity (S), temperature (T), and preformed $O_2$, $NO_3^-$ and $PO_4^{3-}$ , following the approach of Bernardello et al. (2014).

$$
\begin{aligned}
alk_{pre} = &(a_0 + a_1 \cdot S' + a_2 \cdot T' + a_3 \cdot O_{2_{pre}} + a_4 \cdot NO_{3_{pre}}^- + a_5 \cdot PO_{4_{pre}}^{3-}) \cdot NAtl \\
&+ (b_0 + b_1 \cdot S' + b_2 \cdot T' + b_3 \cdot O_{2_{pre}} + b_4 \cdot NO_{3_{pre}}^- + b_5 \cdot PO_{4_{pre}}^{3-}) \cdot SO \\
&+ (c_0 + c_1 \cdot S' + c_2 \cdot T' + c_3 \cdot O_{2_{pre}} + c_4 \cdot NO_{3_{pre}}^- + c_5 \cdot PO_{4_{pre}}^{3-}) \cdot (1 - SO - NAtl)
\end{aligned}
\tag{A5}
$$

where $S' = S - 35$, $T' = T - 20^o C$, the $a_i$ are determined by a regression in the surface North Atlantic, the $b_i$ for the SO, and the $c_i$ using the model output elsewhere in the surface. The tracers SO and NAtl are set to 1 in the surface Southern Ocean (south of $30^o S$) and the North Atlantic (north of $30^o N$), respectively, and are conservatively mixed into the ocean interior. This parametrization induces an uncertainty on the order of 1 $\mu mol/kg$ in globally averaged $DIC_{dis}$ (see fig. A1). As discussed above, however, this is small compared to the signal seen over all simulations.

Finally, $DIC_{dis}$ has been back-calculated from the model output as a residual.

$$DIC_{dis} = DIC - DIC_{sat} - DIC_{soft} - DIC_{carb}$$

*Author contributions.* E. D. Galbraith conducted the model simulations, S. Eggleston performed the analysis, and both contributed to writing the manuscript.

*Competing interests.* The authors declare that they have no conflict of interest.

*Acknowledgements.* The authors would like to thank R. Bernardello for very helpful discussion. S. Eggleston  was funded by a fellowship from the Swiss National Science Foundation. E. D. Galbraith acknowledges computing support from the Canadian Foundation for Innovation and Compute Canada, and financial support from the Spanish Ministry of Economy and Competitiveness, through the María de Maeztu Programme for Centres/Units of Excellence in R&D (MDM-2015-0552).

[revised manuscript text omitted]

$$\mathrm{DIC}_{dis_{global}} + \mathrm{DIC}_{soft_{global}} \approx r_{C:N} \cdot \left(\mathrm{NO3}_{global} + \mathrm{NO3}_{den,global}\right) + \sum_{i=1}^{n} f_i \cdot \left(\mathrm{DIC}_{dis_i} - r_{C:N} \cdot \mathrm{NO3}_{pre_i}\right)$$

$_{dis}$ at steady-state can be estimated fairly robustly as the result of a simple mass balance of the relevant parameters in the most important ventilating water masses. Here, we take into account upper-ocean water masses (above 1 km) formed in the North Pacific, North Atlantic, and Southern Ocean, and deep water masses formed in the North Atlantic and Southern Ocean. In each plot, the full model output is shown on the $x$-axis and the result of the mass balance approximation on the $y$-axis. Orange and blue symbols represent high and low obliquity scenarios, respectively; triangles pointing upward and downward represent greater northern and southern hemisphere seasonality or precession 270° and 90°, respectively; outlines are scenarios with  LGM ice sheets; light shading indicates scenarios with LGM ice sheet topography but PI albedo. The size of the symbols corresponds to the SAT. The purple  square represents  pre-industrial  simulation (run 41), and  red  boxes indicate Fe fertlization simulations (runs 37 – 40).

[Figure]

**Figure A1.** Shown is the difference between the exact $DIC_{dis}$ surface field  $\mu mol\ kg^{-1}$, where $DIC_{sat}$ has been calculated using the surface alkalinity ($alk[z=0] = alk_{pre}[z=0]$) and $DIC_{soft}[z=0] = DIC_{carb}[z=0] = 0$. Differences are shown for (a) LGM; (b) PI; (c) WF.

**Table 1.** Simulation overview. A total of 44 simulations were analyzed with varying radiative forcing (RF), obliquity, precession, ice sheets (PI = pre-industrial; LGM = Last Glacial Maximum reconstruction; LGM* = topography of LGM ice sheets but with PI albedo), and with and without iron fertilization. Runs 1 – 40 are described by Galbraith and de Lavergne (2018). Runs 43 and 44 and identical to 41 and 42 but the remineralization rate of sinking organic matter is reduced by 25%.

| run | RF in $p$CO₂ equivalents (ppm) | obliquity | precession | IS | Fe | remin |
|------|-------------------------------|-----------|------------|------|-----|-------|
| 1-24 | 180, 220, 270, 405, 607, 911 | 22º, 24.5º | 90º, 270º | PI | | |
| 25-28 | 220 | 22º, 24.5º | 90º, 270º | LGM | | |
| 29-32 | 180 | 22º, 24.5º | 90º, 270º | LGM* | | |
| 33-36 | 180 | 22º, 24.5º | 90º, 270º | LGM | | |
| 37-40 | 180 | 22º, 24.5º | 90º, 270º | LGM | X | |
| 41 | 270 | 23.4º | 102.9º | PI | | |
| 42 | 270 | 23.4º | 102.9º | PI | X | |
| 43 | 270 | 23.4º | 102.9º | PI | | 75% |
| 44 | 270 | 23.4º | 102.9º | PI | X | 75% |